# Macrophage network dynamics depend on haptokinesis for optimal local surveillance

**Neil Paterson[1,2,3], Tim Lämmermann[1]\***

[1]Max Planck Institute of Immunobiology and Epigenetics, Freiburg, Germany; [2]International Max Planck Research School for Immunobiology, Epigenetics and Metabolism (IMPRS-IEM), Freiburg, Germany; [3]Faculty of Biology, University of Freiburg, Freiburg, Germany

**Abstract** Macrophages are key immune cells with important roles for tissue surveillance in almost all mammalian organs. Cellular networks made up of many individual macrophages allow for optimal removal of dead cell material and pathogens in tissues. However, the critical determinants that underlie these population responses have not been systematically studied. Here, we investigated how cell shape and the motility of individual cells influences macrophage network responses in 3D culture settings and in mouse tissues. We show that surveying macrophage populations can tolerate lowered actomyosin contractility, but cannot easily compensate for a lack of integrin-mediated adhesion. Although integrins were dispensable for macrophage chemotactic responses, they were crucial to control cell movement and protrusiveness for optimal surveillance by a macrophage population. Our study reveals that β1 integrins are important for maintaining macrophage shape and network sampling efficiency in mammalian tissues, and sets macrophage motility strategies apart from the integrin-independent 3D migration modes of many other immune cell subsets.

**\*For correspondence:** laemmermann@ie-freiburg.mpg.de

**Competing interest:** The authors declare that no competing interests exist.

## Editor's evaluation

You have demonstrated that macrophages utilise integrins for tissue surveillance and network formation in a manner distinct from other leukocytes types. You have put this in context with major mechanisms for forming F-actin based protrusions including Arp2/3 dependent branched F-actin. Your work will be of great interest to immunologists, cell biologies and tissue engineers.

## Introduction

Macrophages are multifunctional immune cells that populate practically all tissues in the body where they play important roles in tissue homeostasis, organ development, inflammation, metabolic adaptation, tumor development, and host defense against pathogens (*Okabe and Medzhitov, 2016*; *Wood and Martin, 2017*). As professional phagocytic cells, one of the major homeostatic functions of macrophages, in both invertebrates and vertebrates, is the removal of dead cell corpses and tissue debris (efferocytosis) (*Cox et al., 2021*; *Wood and Martin, 2017*). While the phases of corpse recognition (find-me), uptake (eat-me), and digestion (digest-me) are molecularly well described in the context of individually responding cells (*Davidson and Wood, 2020b*; *Elliott and Ravichandran, 2016*; *Rothlin et al., 2021*), much less is known about the role of group dynamics in this process. It is well established that macrophages distribute in numerous mammalian tissues to form large networks of many individual cells (*Dawson et al., 2020*; *Freitas-Lopes et al., 2017*; *Gordon et al., 2014*; *Honda et al., 2020*; *Jain and Weninger, 2013*; *Nicolás-Ávila et al., 2020*; *Stolp et al., 2020*; *Uderhardt et al.,*

**eLife digest** Macrophages are immune cells in the body that remove dying cells and debris from tissues. They live in almost all the body's organs, surveilling for signs of infection and destroying microbes. They also migrate to wound sites, where they can eliminate foreign particles and stop microbes from entering the body.

To perform their surveillance role, macrophages need to work together as a team. They form a network, coordinating their movements to optimise the removal of particles and dead cells. How this happens is something of a mystery. As individuals, cells travel through tissues using a balance of several activities: they change their shape, they contract and relax, and they grab hold of their surroundings using proteins called integrins. It is thought that the choice between these types of movement may affect the rest of the network.

To investigate, Paterson and Lämmermann genetically engineered mouse macrophages grown in the laboratory so they would not produce working integrins. These macrophages were able to contract and relax, but they could not attach to the proteins in the structures they were exploring.

Paterson and Lämmermann then placed these macrophages in gels studded with proteins that mimic a biological matrix to observe their behaviour. When these macrophages were exposed to the chemicals that indicate the presence of a wound, they moved normally, changing shape and contracting and relaxing. Paterson and Lämmermann confirmed this normal behaviour for macrophages moving to sites of injuries in the tissue of living mice. However, when it came to surveillance, the macrophages' abilities were seriously diminished, and they were unable to form an effective network to take up particles and dead cells.

This work sheds light on how the movement of individual cells affects the entire immune surveillance network. A deeper understanding could lead to new insights into how to prevent inflammation. The next step is to map macrophage networks in healthy and diseased tissues to understand how cell movement affects surveillance under different conditions.

*2019*). However, the population aspect of macrophage efferocytosis in tissues has so far only received little attention and the single-cell parameters that critically determine the efferocytic capacity of a whole macrophage population are only poorly understood. In particular, it remains unclear how the cytoskeletal control of single-cell shape and movement influences the efferocytic capacity of mammalian macrophage networks.

Cell shape and cell motility are determined by the balanced interplay of three components: actin polymerization, actomyosin contraction, and adhesion to the extracellular environment (*Bodor et al., 2020*). Leukocyte migration in three-dimensional (3D) interstitial spaces is considered flexible and adaptive, with many immune cells switching to alternate migration modes upon perturbations of any of these three components (*Lämmermann and Sixt, 2009*). Our current view on interstitial leukocyte motility is still largely influenced by studies with fast-migrating immune cells (dendritic cells, neutrophils, lymphocytes) that traffic between parenchyma and vasculature of mouse tissues (*Lämmermann and Germain, 2014*). Previous work on these cell types has highlighted that leukocyte motility outside the vasculature relies almost exclusively on cell shape changes driven by the actomyosin cytoskeleton (*Lämmermann et al., 2013*; *Lämmermann et al., 2008*; *Woolf et al., 2007*). This migration mode, commonly referred to as amoeboid migration, is independent from integrin adhesion receptors and strong adhesive interactions with the tissue environment (*Paluch et al., 2016*; *Reversat et al., 2020*). Macrophages, as archetypes of tissue-resident immune cells, appear to contrast most other leukocytes. Macrophages in zebrafish larvae (*Barros-Becker et al., 2017*) and human monocyte-derived macrophages (hMDMs) invading into 3D matrigels (*Van Goethem et al., 2011*; *Van Goethem et al., 2010*) move with elongated morphology at lower speeds and show adhesion structures containing heterodimeric integrin receptors (*Hynes, 2002*). This mode of locomotion is best described as a mesenchymal-like migration and rather resembles the movement patterns of fibroblasts and other non-immune cell types (*Yamada and Sixt, 2019*). Macrophages in mice also protrude elongated processes for tissue surveillance in almost all organs. Moreover, they are well known as very adhesive cell type with a broad range of different integrin adhesion receptors on their surface (*Ley et al., 2016*). However, we still have very limited knowledge on the functional contribution of integrins to

macrophage shape, positioning and migration in 3D interstitial spaces of murine tissues. Studies of *Drosophila* macrophages, so called hemocytes, provide currently the best insight into this question. Mutating the mainβPS integrin in *Drosophila* causes hemocyte migration deficits in embryos (*Comber et al., 2013*) and late pupal stages (*Moreira et al., 2013*), but a direct comparison between *Drosophila* and mouse macrophages is problematic. Hemocytes move in highly confined, fluid-filled spaces, where only over time these macrophages together with other cells deposit extracellular matrix (ECM) (*Matsubayashi et al., 2017*; *Sánchez-Sánchez et al., 2017*). Thus, it remains unclear how these findings relate to macrophage behavior in geometrically complex, often matrix-rich interstitial spaces of mammalian tissues. Here, we systematically address how lack of integrin functionality influences macrophage motility in mouse tissues and 3D in vitro matrices, and how these cells adapt to a loss of adhesiveness. As the central point of this study, we investigate how the cell shape and the motility mode of individual cells influence the efferocytic efficiency of macrophage networks, and how perturbations on the single-cell level may be compensated in a sampling phagocyte population.

## Results

### Haptokinetic random motility of macrophages in 3D matrices

Macrophages distribute homogeneously as cellular networks in most mouse tissues, as exemplified by tissue-resident macrophages of the brain-surrounding dura mater (*Figure 1A*). Studying network dynamics and migration of slow-migrating macrophages by two-photon intravital microscopy (2P-IVM) is however challenging and often limited to only a few hours. To overcome this restriction, we established an in vitro platform for the microscopic observation of macrophage network dynamics over 24 hr and longer (*Figure 1B* and *Figure 1—video 1*). We used primary mouse bone marrow-derived macrophages (BMDMs) (*Weischenfeldt and Porse, 2008*; *Zajd et al., 2020*), which were embedded into 3D matrigel. By combining this system with video-based brightfield microscopy, we monitored migration dynamics in macrophage populations over 24–30 hr and found that individual cells moved with mesenchymal-like elongated shapes at average speeds of ~0.6 µm/min (*Figure 1C–E*). Treatment of BMDMs with the F-actin-disrupting drug cytochalasin D, the Rho-associated kinase (ROCK) inhibitor Y27632 or the non-muscle myosin II inhibitor blebbistatin revealed an essential requirement of actin dynamics and an important role of actomyosin contraction for macrophage random migration (*Figure 1C–E*, *Figure 1—figure supplement 1*). Treatment of BMDMs with the Arp2/3 complex inhibitor CK-666 caused cell rounding and loss of prominent mesenchymal protrusions in the majority of macrophages (*Figure 1—video 2*). This resulted in a significant reduction in the average speed, supporting an important role of dendritic actin filament networks for 3D macrophage random migration (*Figure 1—figure supplement 1* and *Figure 1—video 2*). To address the functional role of integrin-mediated adhesion, the third component determining cell migration, for macrophage 3D motility, we used different mouse crosses to generate BMDMs without functional high-affinity integrins (*Tln1*⁻ᐟ⁻) or without cell surface integrin heterodimers of the β2 family (*Itgb2*⁻ᐟ⁻) or β1 family (*Itgb1*⁻ᐟ⁻) (*Figure 1—figure supplements 2 and 3*). *Tln1*⁻ᐟ⁻ BMDMs, which are depleted of talin, a crucial interactor with integrin cytoplasmic domains for integrin activation (*Calderwood and Ginsberg, 2003*), showed roundish, amoeboid-like morphologies and severely impaired random migration (*Figure 1F–I* and *Figure 1—video 2*). Confocal fluorescence microscopy of Lifeact-GFP expressing BMDMs in 3D matrices clearly revealed that *Tln1*⁻ᐟ⁻ cells were missing the prominent branched protrusions that gave WT macrophages their mesenchymal-like cell shape (*Figure 1K*). *Itgb2*⁻ᐟ⁻ BMDMs lack β2 integrins on the cell surface, including the strongly expressed heterodimer αMβ2 (Mac-1), a characteristic macrophage cell surface protein with promiscuous binding properties to more than 30 non-protein and protein molecules, including several ECM components (*Yakubenko et al., 2002*; *Figure 1—figure supplement 3*). Surprisingly, ITGB2 deficiency did not result in cell shape changes or migration deficiencies (*Figure 1F–I*). However, *Itgb1*⁻ᐟ⁻ BMDMs that lack the important ECM-binding heterodimers α5β1 and α6β1 on their cell surface, copied the morphology and migration phenotype of *Tln1*⁻ᐟ⁻ BMDMs (*Figure 1F–I*; *Figure 1—figure supplement 3*). Thus, our results identify a crucial role of β1 integrins for the mesenchymal-like movement of macrophages, highlighting the substrate-dependent (haptokinetic) nature of 3D macrophage random migration. Loss of integrin functionality and the associated switch from mesenchymal-like to amoeboid morphology results in severely impaired 3D motility, which macrophages cannot compensate in contrast to many other immune cell types.

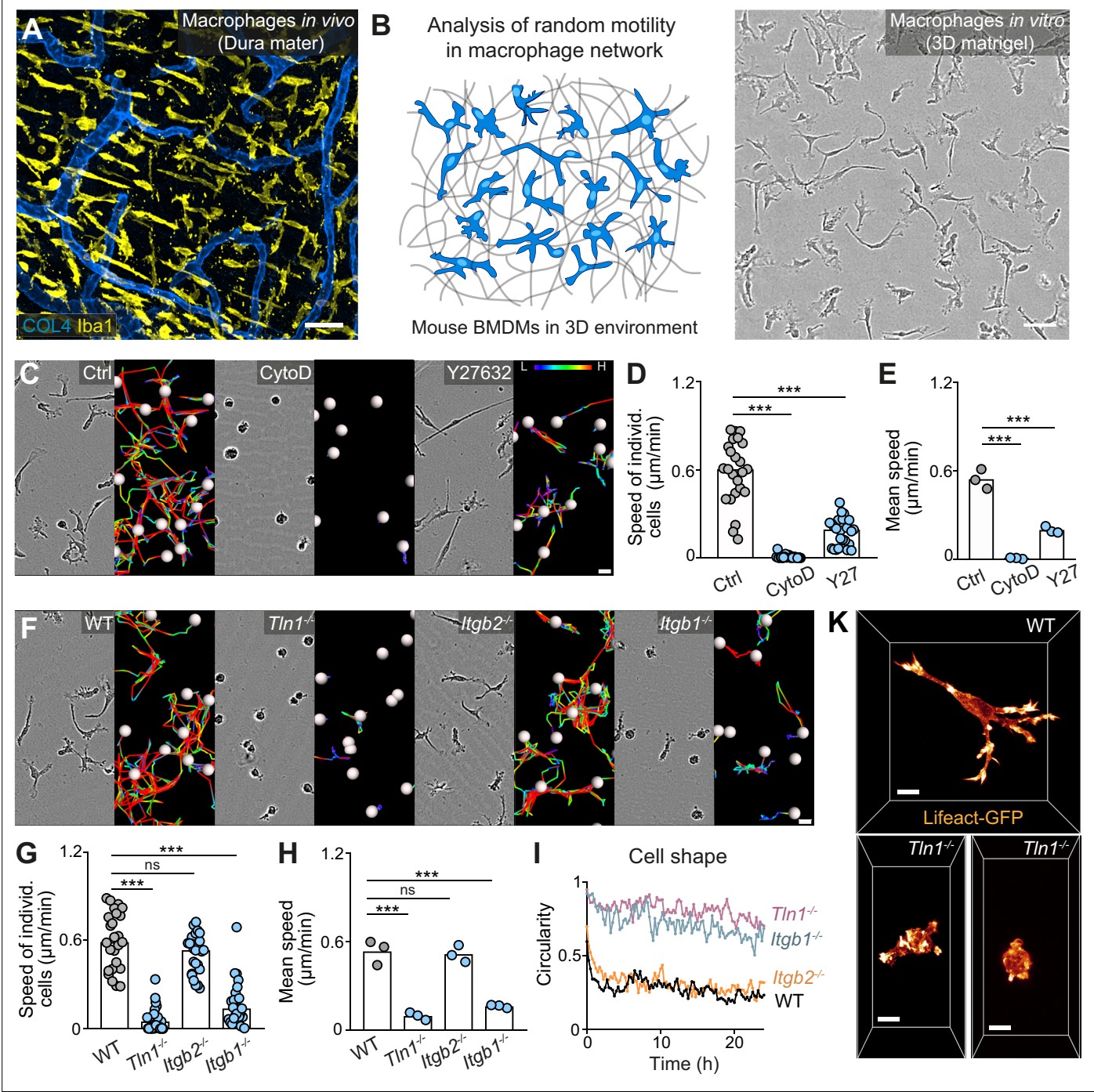

**Figure 1.** Haptokinetic random motility of macrophages in three-dimensional (3D) matrices. (**A**) Representative macrophage network in adult mouse tissue. Immunofluorescence staining of a dura mater whole mount preparation, showing macrophages (yellow) and blood vasculature (blue). COL4: collagen IV. (**B**) Scheme (left) and brightfield image (right) for studying macrophage network dynamics in 3D in vitro matrices. (C–E) Analysis of bone marrow-derived macrophage (BMDM) random motility in the presence of cytochalasin D (CytoD) or Y27632. (**C**) Representative cell morphologies (brightfield microscopy) and pseudo-colored tracks (displacement delta length: L(low) = 0, H(high) = 15) over 24 hr. (**D**) Individual cell speeds from one independent experiment (dots represent randomly chosen cells per condition, N = 25), and (**E**) mean speed values calculated from three biological replicates (n = 3). (F–H) Analysis of BMDM random motility upon genetic interference with integrin functionality, including (**F**) cell morphologies and tracks over 24 hr, (**G**) individual cell speeds from one independent experiment (N = 25), and (**H**) mean speed values calculated from three biological replicates (n = 3). (**I**) Graphical analysis of cell shape at 15 min time intervals over 24 hr for integrin-mutant BMDMs. Dots are mean values of N = 5 randomly chosen cells per genotype. A circularity value of 1 equals a perfectly circular cell. (**K**) Confocal live-cell microscopy of Lifeact-GFP expressing WT or $Tln1^{-/-}$ BMDMs in 3D matrigel. Bars in graphs: median (**D, G**), mean (**E, H**). Statistical tests: ***p ≤ 0.001, Dunn's multiple comparison (post hoc

*Figure 1 continued on next page*

Figure 1 continued

Kruskal-Wallis test) (**D, G**); ***p ≤ 0.001, Dunnett's multiple comparison (post hoc analysis of variance [ANOVA]) (**E, H**). Scale bars: 50 µm (**A, B**), 10 µm (**K**), 20 µm (**C, F**).

The online version of this article includes the following video, source data, and figure supplement(s) for figure 1:

**Source data 1.** Numerical data for the graph in *Figure 1D*.

**Source data 2.** Numerical data for the graph in *Figure 1E*.

**Source data 3.** Numerical data for the graph in *Figure 1G*.

**Source data 4.** Numerical data for the graph in *Figure 1H*.

**Source data 5.** Numerical data for the graph in *Figure 1I*.

**Figure supplement 1.** Actomyosin contraction and dendritic actin networks regulate random motility of macrophages in three-dimensional (3D) matrices.

**Figure supplement 1—source data 1.** Numerical data for the graph in *Figure 1—figure supplement 1B*.

**Figure supplement 1—source data 2.** Numerical data for the graph in *Figure 1—figure supplement 1D*.

**Figure supplement 1—source data 3.** Numerical data for the graph in *Figure 1—figure supplement 1E*.

**Figure supplement 2.** Characterization of mouse bone marrow-derived macrophages (BMDMs) with impaired integrin functionality.

**Figure supplement 2—source data 1.** Non-annotated Western blot for *Figure 1—figure supplement 2B*.

**Figure supplement 2—source data 2.** Annotated Western blot for *Figure 1—figure supplement 2B*.

**Figure supplement 3.** Characterization of mouse bone marrow-derived macrophages (BMDMs) with impaired integrin functionality.

**Figure 1—video 1.** Random motility of macrophages in three-dimensional (3D) matrices.
https://elifesciences.org/articles/75354/figures#fig1video1

**Figure 1—video 2.** Talin-1 and β1 integrins control the mesenchymal shape and random motility of macrophages in matrigel.
https://elifesciences.org/articles/75354/figures#fig1video2

## β1 integrins determine the mesenchymal-like shape of macrophages in mouse tissues

To corroborate the importance of our in vitro findings for living tissues, we investigated tissue-resident macrophages of mice with conditional *Itgb1* deletion in hematopoietic cells. This genetic approach allowed the efficient depletion of ITGB1 in hematopoietic stem cells and thus targeted also endogenous macrophages of different organs and ontogeny. Other genetic strategies (e.g. *Lyz2^CRE^*, *Cx3cr1^CRE^*) resulted in partial targeting of macrophage subsets or incomplete protein depletion, which did not provide conclusive in vivo results (data not shown). When we analyzed endogenous macrophage networks in several ECM-rich tissues by immunofluorescence analysis, the comparison of *Vav1-iCre Itgb1^fl/fl^* mice with littermate controls provided a clear morphological phenotype. In agreement with our findings from 3D matrigels, ITGB1 depletion caused macrophages in the interstitial spaces of the skin dermis (*Figure 2A*), the splenic red pulp (*Figure 2B*), and in the sinusoidal spaces of the liver (*Figure 2C*) to adopt a roundish, amoeboid-like cell shape. In contrast, macrophages in tissues of *Itgb2^−/−^* mice retained their mesenchymal-like morphologies (*Figure 2—figure supplement 1*). Thus, our results confirm the crucial role of β1 integrins for defining the mesenchymal-like shape of endogenous macrophages in several mouse tissues.

## Integrin-independent macrophage movement during chemotactic responses

As external guidance signals can induce cell polarization and directed migration, we next examined the chemotactic migration response of macrophages. We embedded BMDMs in 3D matrigel scaffolds and followed their directed migration along a gradient of the chemoattractant complement factor 5a (C5a) over 24 hr (*Figure 3A*). We then assessed the contribution of actin dynamics, actomyosin contraction, and integrin function to this process (*Figure 3B and C*; *Figure 3—figure supplement 1*). The effects of cytochalasin D and Y27632 treatment on chemotactic macrophage migration were comparable to our previous results on random motility (*Figure 1C–E*), showing an essential requirement for actin dynamics and an important role for actomyosin contraction (*Figure 3B, D and E*). CK-666 treatment did not impair BMDM chemotaxis in 3D matrigel (*Figure 3—figure supplement 1*), which is in agreement with previous studies showing that Arp2/3 complex blockade rather

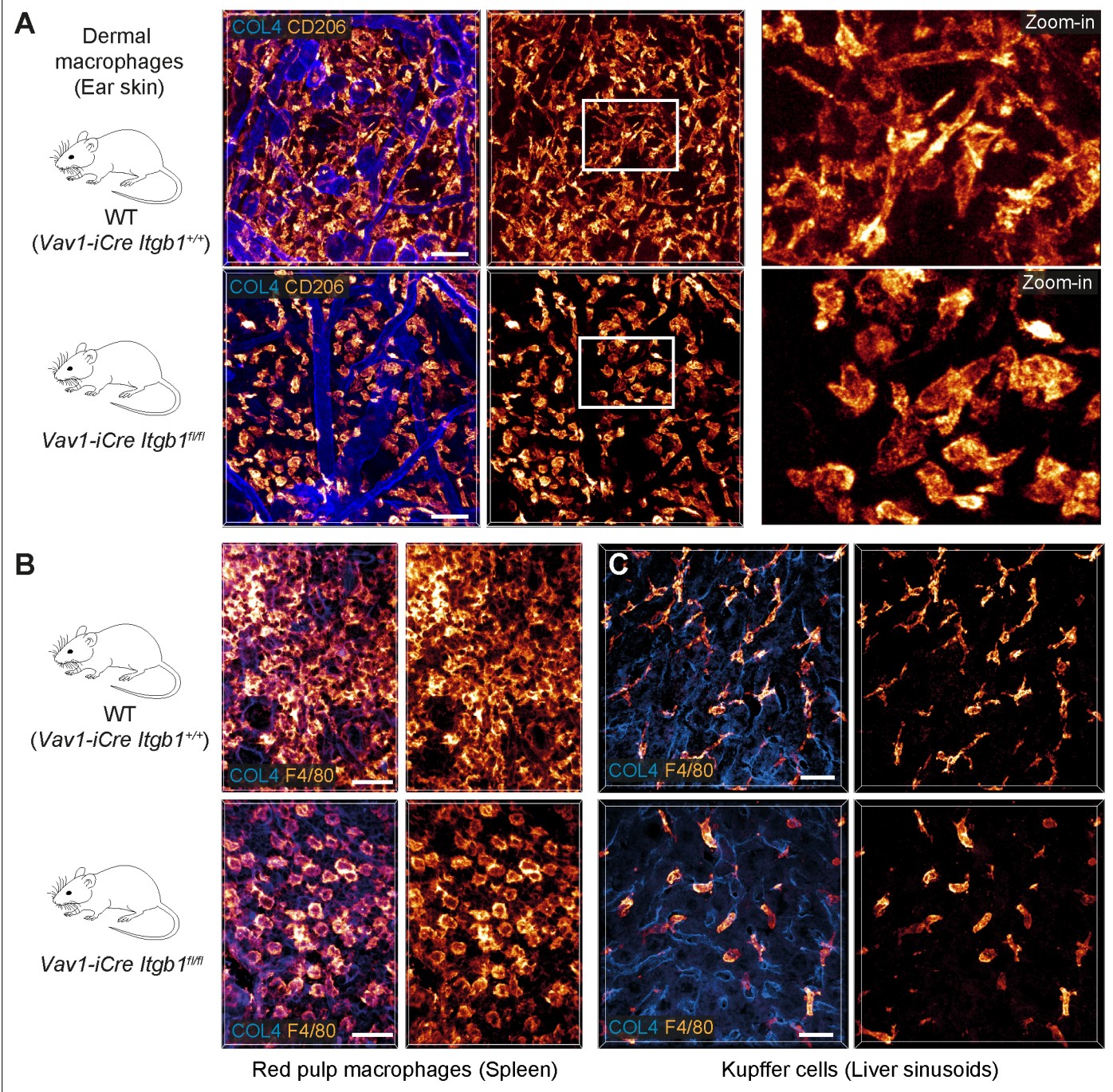

**Figure 2.** β1 integrins define the mesenchymal shape of macrophages in mouse tissues. (**A–C**) Comparative analysis of ear skin dermis (**A**), spleen (**B**) and liver (**C**) tissues of adult *Vav1-iCre*[+/−] *Itgb1*[fl/fl] mice and littermate controls. Endogenous macrophage subsets were detected with immuno-stainings against CD206 (**A**) and F4/80 (**B, C**) and fluorescence signal intensities displayed as glow heatmap color. Collagen IV (COL4)-expressing basement membrane (**A, C**) or reticular network (**B**) structures are also displayed (blue). All images are projections of several confocal z-planes. Scale bars: 50 μm (**A**), 30 μm (**B, C**).

The online version of this article includes the following figure supplement(s) for figure 2:

**Figure supplement 1.** β2 integrins do not contribute to the mesenchymal shape of macrophages in mouse tissues.

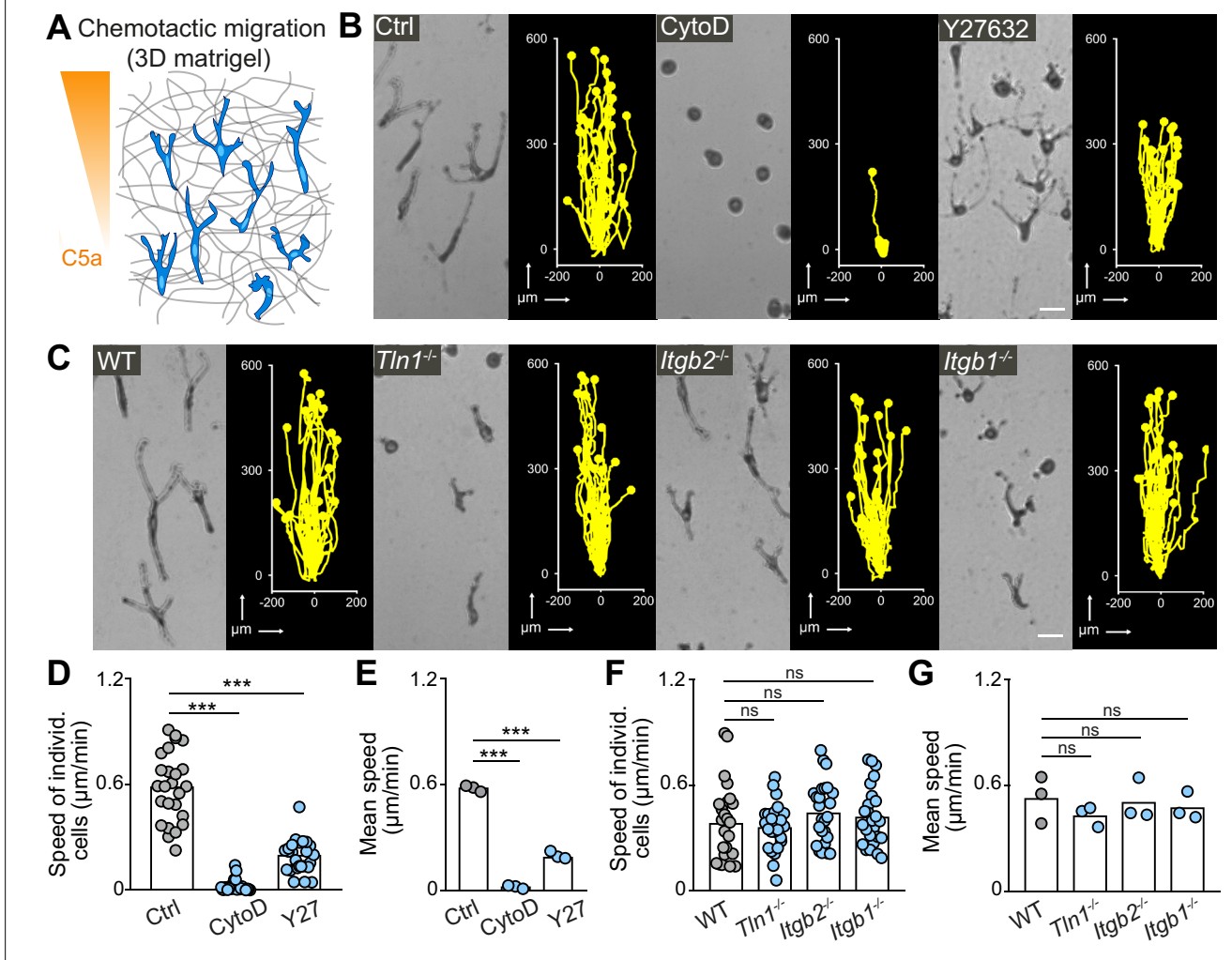

**Figure 3.** Integrin-independent three-dimensional (3D) macrophage movement during chemotactic responses. (**A**) Scheme for studying chemotactic macrophage migration toward C5a gradients in 3D in vitro matrices. (**B, C**) Representative cell morphologies (brightfield microscopy) and tracks over 24 hr of chemotaxing bone marrow-derived macrophages (BMDMs) (**B**) in the presence of cytochalasin D (CytoD) and Y27632, and (**C**) upon genetic interference with integrin functionality. Scale bars: 25 µm. (**D–G**) Analysis of BMDM chemotactic migration, including (**D,F**) individual cell speeds from one independent experiment (dots represent randomly chosen cells per condition, N = 25), and (**F,G**) mean speed values of three biological replicates (n = 3). Bars in graphs: median (**D–F**), mean (**E–G**). Statistical tests: ***p ≤ 0.001, Dunn's multiple comparison (post hoc Kruskal-Wallis test) (**D**); ***p ≤ 0.001, Dunnett's multiple comparison (post hoc analysis of variance [ANOVA]) (**E–G**).

The online version of this article includes the following video, source data, and figure supplement(s) for figure 3:

**Source data 1.** Numerical data for the graph in *Figure 3D*.

**Source data 2.** Numerical data for the graph in *Figure 3E*.

**Source data 3.** Numerical data for the graph in *Figure 3F*.

**Source data 4.** Numerical data for the graph in *Figure 3G*.

**Figure supplement 1.** Arp2/3 complex-mediated dendritic actin networks are dispensable for macrophage chemotaxis in three-dimensional (3D) matrigel.

**Figure supplement 1—source data 1.** Numerical data for the graph in *Figure 3—figure supplement 1B*.

**Figure supplement 2.** Track straightness and C5aR1 expression are unaltered in chemotaxing bone marrow-derived macrophages (BMDMs) with impaired integrin functionality.

**Figure 3—video 1.** Integrin-independent chemotactic movement of macrophages in matrigel.

https://elifesciences.org/articles/75354/figures#fig3video1

**Figure supplement 2—source data 1.** Numerical data for the graph in *Figure 3—figure supplement 2A*.

**Figure supplement 2—source data 2.** Numerical data for the graph in *Figure 3—figure supplement 2C*.

increases than decreases migration speed in several cell types (*Asokan et al., 2014*; *Dimchev et al., 2021*; *Georgantzoglou et al., 2021*; *Leithner et al., 2016*; *Moreau et al., 2015*; *Rotty et al., 2017*; *Vargas et al., 2016*; *Wu et al., 2012*). Strikingly, the dependency on integrin function was markedly different between random and chemotactic macrophage migration. In contrast to random motility, *Tln1*⁻/⁻ and *Itgb1*⁻/⁻ BMDMs, which had adopted more roundish and amoeboid-like shapes, moved at comparable average speeds to WT and *Itgb2*⁻/⁻ BMDMs, which migrated with very elongated and mesenchymal-like morphologies along the C5a gradient (*Figure 3C, F and G* and *Figure 3—video 1*). Track straightness and cell surface expression of C5aR1 were comparable between all gene variants (*Figure 3—figure supplement 2*). Y27632 treatment impaired the chemotactic migration of *Tln1*⁻/⁻ BMDMs, supporting an important role of actomyosin contractility as amoeboid protrusive force for integrin-independent macrophage migration (*Figure 3—figure supplement 2*). Thus, chemotactic guidance cues can overcome the migration deficit of adhesion-deficient macrophages and induce productive amoeboid-like, integrin-independent macrophage movement.

## Amoeboid-like macrophages still perform chemotactic migration in mouse tissue

To confirm our findings in vivo, we chose to investigate the chemotactic response of tissue-resident macrophages to laser-induced wounds in the mouse dermis (*Figure 4A*). Previous intravital imaging studies in mice demonstrated wound attractants to induce chemotactic responses of several myeloid cell types, mostly neutrophils and tissue-resident macrophages (*Lämmermann et al., 2013*; *Uderhardt et al., 2019*). We crossed *Vav1-iCre Itgb1*ᶠˡ/ᶠˡ mice with lysozyme M-GFP (*Lyz2*ᴳᶠᴾ) knock-in reporter mice to visualize dermal myeloid cells by 2P-IVM. In agreement with our immunofluorescence analysis of ear skin whole mount tissues (*Figure 2A*), 2P-IVM of GFP-positive macrophages in the unchallenged ear dermis confirmed that most ITGB1-deficient macrophages lacked the typical multi-protrusive mesenchymal-like phenotype of WT macrophages (*Figure 4—figure supplement 1*). As neutrophils can influence macrophage dynamics at the wound site, we removed them from the blood circulation by administering Anti-Ly6G neutrophil-depleting antibodies. This experimental strategy allowed us to accurately analyze the functional contribution of β1 integrins in the chemotactic wound response of tissue-resident macrophages (*Figure 4A*). In contrast to our in vitro imaging over a whole day, 2P-IVM was limited to 90–120 min. Imaging WT macrophages at high magnification revealed that these cells quickly formed long protrusions toward the tissue lesion, while most cell bodies remained immotile during this short observation period (*Figure 4B* and *Figure 4—video 1*). Although most *Itgb1*⁻/⁻ macrophages displayed rounded morphologies in unchallenged skin, these cells also formed directed protrusions toward the damage site at comparable speeds to WT cells (*Figure 4C and D* and *Figure 4—video 1*). We observed for *Itgb1*⁻/⁻ macrophages a twofold increase in cell body displacement (53% of all analyzed *Itgb1*⁻/⁻ cells, N = 55) in comparison to WT macrophages (24% of all analyzed WT cells, N = 34), which we interpret as a switch to a more amoeboid migration mode (*Figure 4E*; *Figure 4—figure supplement 2* and *Figure 4—video 1*). Thus, our in vivo results confirm the dispensable role of integrins for the chemotactic response of macrophages. They also show that chemotactic cues are sufficient to polarize the macrophage cytoskeleton and support directed integrin-independent 3D protrusive movement. These findings expand our previous results on the chemotactic behavior of fast-migrating immune cells (dendritic cells, neutrophils, B cells) (*Lämmermann et al., 2008*) to a slower migrating tissue-resident immune cell type.

## Two efficient surveillance strategies for macrophage networks

Next, we investigated how cell shape changes and motility modes of individual macrophages influence the surveillance behavior of a whole macrophage network. To address this question, we adapted our 3D in vitro platform and added fluorescent beads with attached phosphatidylserine (PS) to macrophage populations in matrigel (*Figure 5A*). PS on the bead surface acted as an 'eat-me' signal for macrophages (*Segawa and Nagata, 2015*), and a network of 400–500 macrophages was able to almost completely clear gels of a corresponding number of extracellular particles within 24 hr (*Figure 5B*). Several hours after ingestion by macrophages, bead fluorescence was quenched due to the acidic environment of the phagolysosomal system, which could be measured as overall reduction in fluorescence (*Figure 5—figure supplement 1*). Using this system, we set out to understand the cytoskeletal requirements of sampling macrophage populations. Cytochalasin D treatment served

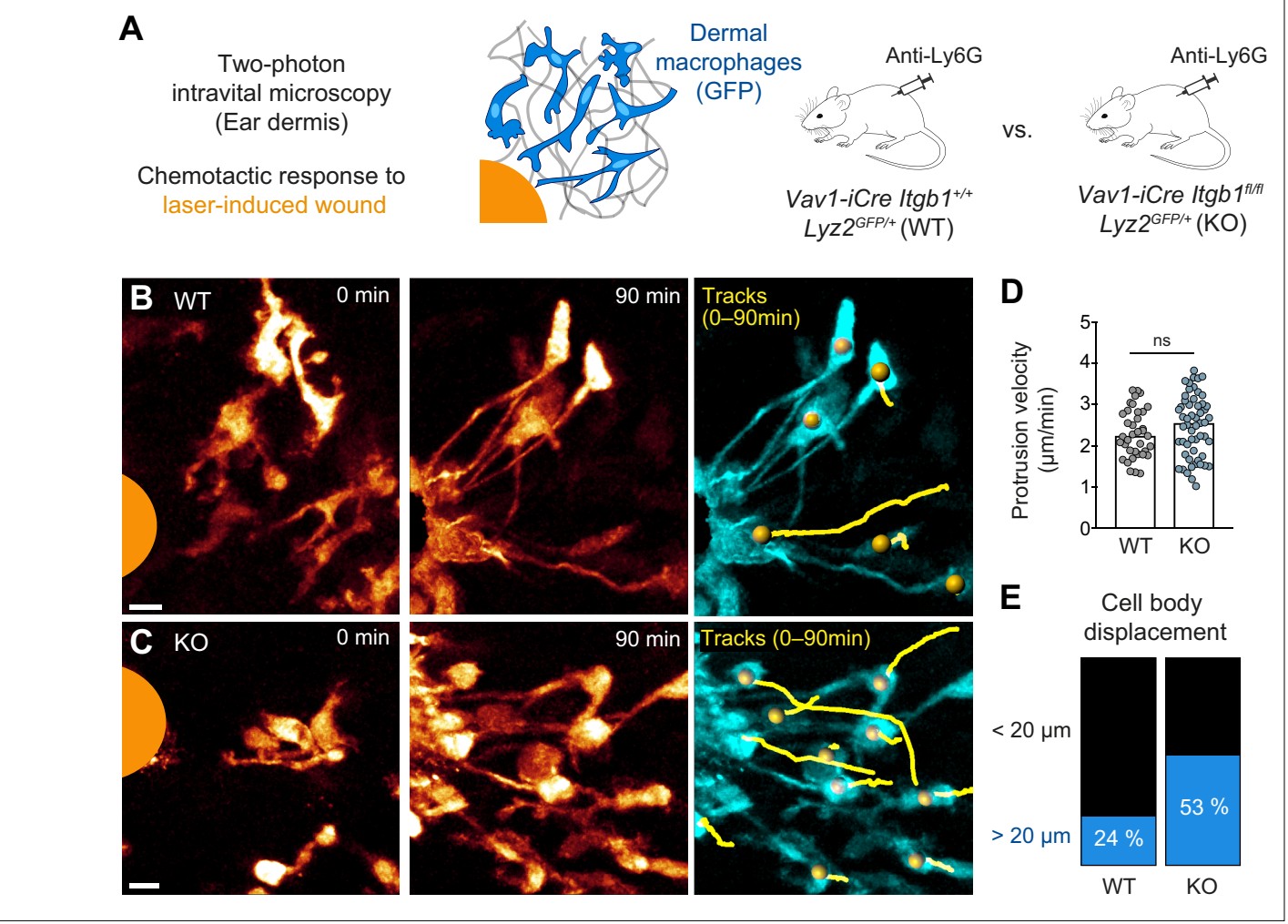

**Figure 4.** Amoeboid-like macrophages still perform chemotactic migration in mouse tissue. (**A**) Scheme for studying the chemotactic response of dermal macrophages to laser-induced tissue injury in mouse ear skin. Two-photon intravital microscopy (2P-IVM) was performed on *Vav1-iCre Itgb1*$^{fl/fl}$ *Lyz2*$^{GFP/+}$ and littermate control mice. Mice were treated with Anti-Ly6G antibody to deplete neutrophils and avoid their presence in imaging field of views. (**B, C**) 2P-IVM images of GFP-expressing dermal macrophages in WT mice (**B**) and conditional *Itgb1*-deficient mice (**C**) at the onset of the wound response and 90 min later. GFP signal is displayed as glow heatmap color. Cell body displacements are shown as yellow tracks. Scale bars: 10 µm. (**D**) Velocity analysis of macrophage protrusions moving toward the tissue lesion. Each dot represents one protrusion (WT: N = 37; KO: N = 55). Values are pooled from n = 3 (WT) and n = 4 (KO) mice; ns: non-significant, *U* test. Bars are median. (**E**) Cell bodies of responding macrophages were tracked and categorized according to displacement (WT: N = 34; KO: N = 55). Values are pooled from n = 3 (WT) and n = 4 (KO) mice.

The online version of this article includes the following video, source data, and figure supplement(s) for figure 4:

**Source data 1.** Numerical data for the graph in *Figure 4D*.

**Figure supplement 1.** Two-photon intravital microscopy of macrophage shapes in unchallenged mouse skin.

**Figure supplement 2.** *Itgb1*-deficient dermal macrophages perform chemotactic migration in mouse skin tissue.

**Figure 4—video 1.** Amoeboid-like macrophages still perform chemotactic responses in mouse tissue.

https://elifesciences.org/articles/75354/figures#fig4video1

as negative control for our analysis, as the complete stalling of migration and protrusion formation inhibited macrophage space exploration and bead uptake over 24 hr (*Figure 5C–E*). In contrast, lowering actomyosin contractility by Y27632 treatment did not impact the sampling efficiency of a macrophage population (*Figure 5C–E*). Although individual BMDMs moved under these conditions at only ~30% of their normal speed (*Figure 1D–E*), their gain in single-cell protrusiveness compensated for this reduction in speed and allowed Y27632-treated macrophage populations equal space exploration and bead uptake as control populations (*Figure 5C and F*; *Figure 5—figure supplement*

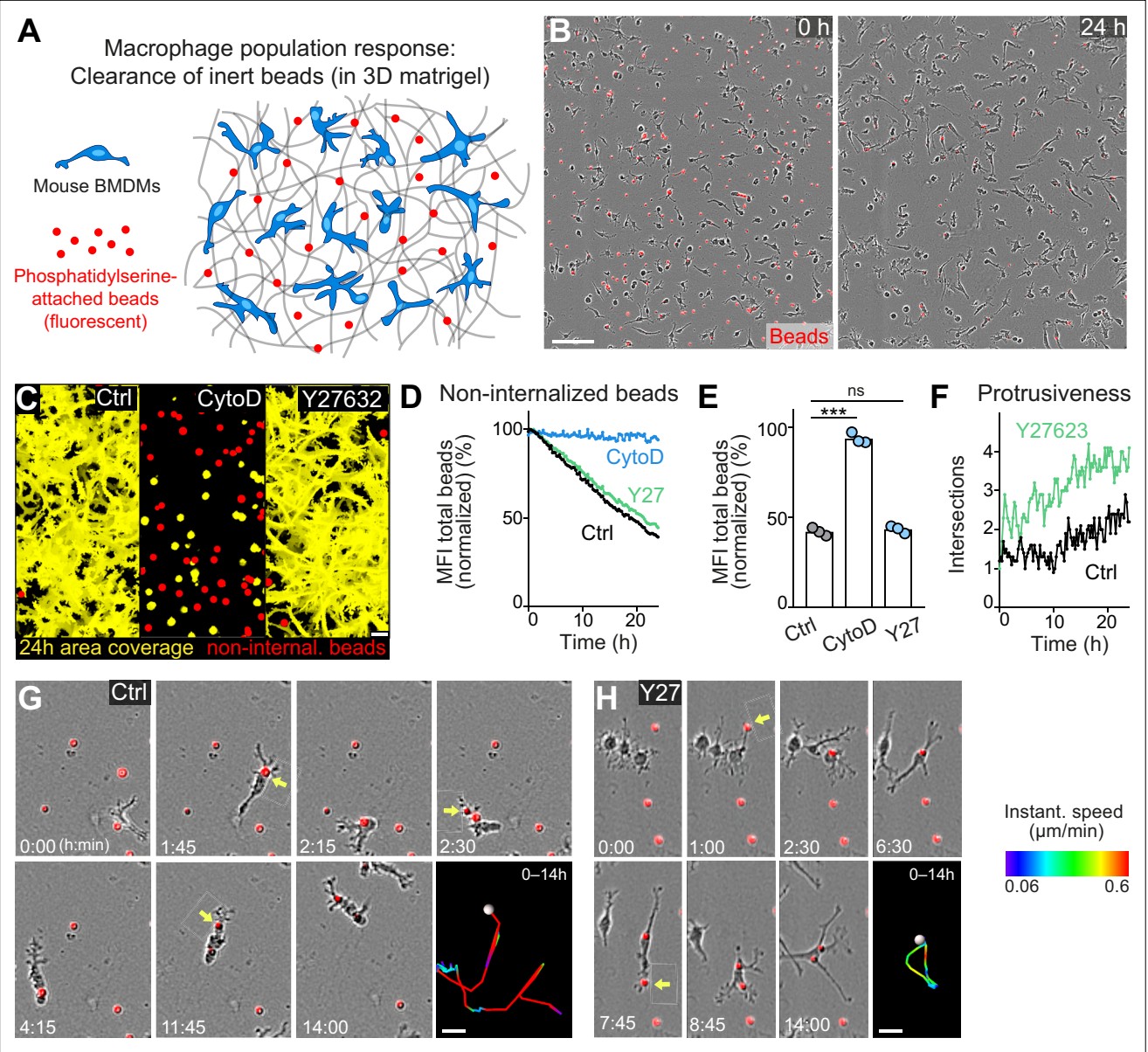

**Figure 5.** Movement and protrusiveness as two sampling strategies for bead removal by macrophage networks. (**A**) Scheme for studying macrophage network surveillance in three-dimensional (3D) in vitro matrices. (**B**) Images of start (0 hr) and endpoint (24 hr) of bead removal by a population of WT bone marrow-derived macrophages (BMDMs) (unstained). Extracellular, fluorescent beads with surface-attached phosphatidylserine (red, 0 hr) were ingested by BMDMs over time. The image shows a quarter of the total imaging field of view. (**C–E**) Analysis of BMDM network sampling activity in the presence of cytochalasin D (CytoD) or Y27632, including (**C**) time projections of macrophage shapes over 24 hr, displayed as total area coverage (yellow) in relation to non-internalized beads (red). Bead sampling by macrophages was measured as mean fluorescence intensity (MFI) decline of bead fluorescence in 15 min intervals over time, presented as (**D**) time-course analysis from one independent experiment (dots in curves are mean values from N = 4–5 technical replicates (separate wells of matrigel)), and as (**E**) 24 hr-mean-values calculated from three biological replicates (n = 3 per genotype). (**F**) Cell protrusiveness of WT and Y27632-treated BMDMs was determined by Sholl analysis for N = 10 randomly chosen cells and presented as mean values at 15 min time intervals over 24 hr. (**G, H**) Time sequences of individual control (**G**) and Y27632-treated (Y27) (**H**) BMDMs, correlating bead sampling and migratory activity. Yellow arrows highlight bead uptake events. Cell tracks over 14 hr are pseudo-colored for instantaneous speed values. All bar graphs display the mean; ***p ≤ 0.001, ns: non-significant; Dunnett's multiple comparison (post hoc analysis of variance [ANOVA]). Scale bars: 100 µm (**B**), 40 µm (**C**), 20 µm (**G, H**).

The online version of this article includes the following video, source data, and figure supplement(s) for figure 5:

**Source data 1.** Numerical data for the graph in *Figure 5D*.

**Source data 2.** Numerical data for the graph in *Figure 5E*.

*Figure 5 continued on next page*

*Figure 5 continued*

**Source data 3.** Numerical data for the graph in *Figure 5F*.

**Figure supplement 1.** Removal of fluorescent phosphatidylserine-attached beads by macrophage networks and individual macrophages.

**Figure supplement 2.** Measurement of cell protrusiveness by Sholl analysis (relates to *Figure 5F*).

**Figure 5—video 1.** Movement and protrusiveness as two sampling strategies for bead removal by macrophage networks.

https://elifesciences.org/articles/75354/figures#fig5video1

*2*). Thus, macrophage networks can use two equally efficient surveillance strategies: (a) surveillance by migration (*Figure 5G* and *Figure 3—video 1*) and (b) surveillance by low-motile protrusion extension (*Figure 5H* and *Figure 5—video 1*). Considering the broad heterogeneity of macrophages in mammalian tissues (*Blériot et al., 2020*), this finding is very relevant and highlights that efficient tissue surveillance can be realized by macrophage subsets with migratory potential, but also by macrophage subsets with sessile, but protrusive behaviors.

## Haptokinesis is required for optimal bead removal by macrophage networks

We then investigated how integrin-dependent haptokinesis influences the sampling efficiency of macrophage networks. We found that integrin-dependent deficits in macrophage random motility (*Figure 1F–I*) translated directly to impaired removal of extracellular particles in matrigel (*Figure 6A–C*). Although *Tln1*$^{-/-}$ and *Itgb1*$^{-/-}$ BMDMs had comparable phagocytic activity to WT controls when macrophages were kept as cell suspensions during the incubation with PS-attached beads (*Figure 6—figure supplement 2*), these mutant BMDMs showed clearly reduced bead internalization in the 3D matrix (*Figure 6B and C*; *Figure 6—figure supplement 1*). Impaired haptokinesis of amoeboid-shaped *Tln1*$^{-/-}$ and *Itgb1*$^{-/-}$ BMDMs impeded efficient space exploration and bead sampling, which was not observed for mesenchymal-like migrating *Itgb2*$^{-/-}$ BMDMs (*Figure 6A–C*; *Figure 6—figure supplement 1*). Live-cell imaging analysis revealed that the rudimentary movement of *Itgb1*$^{-/-}$ BMDMs was sufficient to sample beads in close vicinity to them, but the restricted movement radius prevented bead sampling of larger areas (*Figure 6D* and *Figure 6—video 1*). However, we could rescue this surveillance deficit of *Itgb1*$^{-/-}$ BMDM networks by doubling the cell number in the macrophage population (*Figure 6E and F*). This result appears particularly relevant for physiological mammalian tissues, where the additional recruitment of monocytic cells or macrophages might compensate for insufficient space exploration of a tissue-resident macrophage network.

## Haptokinesis is required for optimal efferocytosis by macrophage networks

The removal of dead cell material is best described as a sequential series of cell biological events divided into 'find-me', 'eat-me', and 'digest-me' phases (*Lemke, 2019*). To realize 'eat-me', individual macrophages are considered to chemotactically respond to 'find-me' signals released from dead cells, a process that involves the formation of directed protrusions and subsequent cell displacement. However, it still remains unresolved which mechanisms guide and coordinate the dynamics of individual cells in macrophage networks where many phagocytes act together, but also compete for dead cell material. Given our disparate findings on integrin-dependent random motility and integrin-independent chemotactic responses of macrophages, we were particularly interested how loss of integrin functionality influences the sampling dynamics of macrophage networks. To study the efferocytosis response of macrophage populations, we added aged, fluorescently labeled mouse neutrophils to BMDM networks (*Figure 7A*). Aged neutrophils underwent cell death over time, and we used pHrodo-Red as fluorescent dye to label them. Efferocytosis was extremely efficient and a network of 300–400 macrophages was able to remove 500–700 dead neutrophils within 24–30 hr (*Figure 7B* and *Figure 7—video 1*). We observed that pHrodo fluorescence after the reported increase shortly after ingestion vanished inside macrophages several hours after neutrophil uptake, probably due to digestion of the corpse. We used this to measure the efferocytosis efficiency of BMDM networks as fluorescence decline, which we quantified by using two independent analysis programs (*Figure 7C*; *Figure 7—figure supplement 1*). Microscopic observation of individual WT BMDMs revealed a spectrum of mesenchymal-like movement behaviors that supported efficient efferocytosis, including

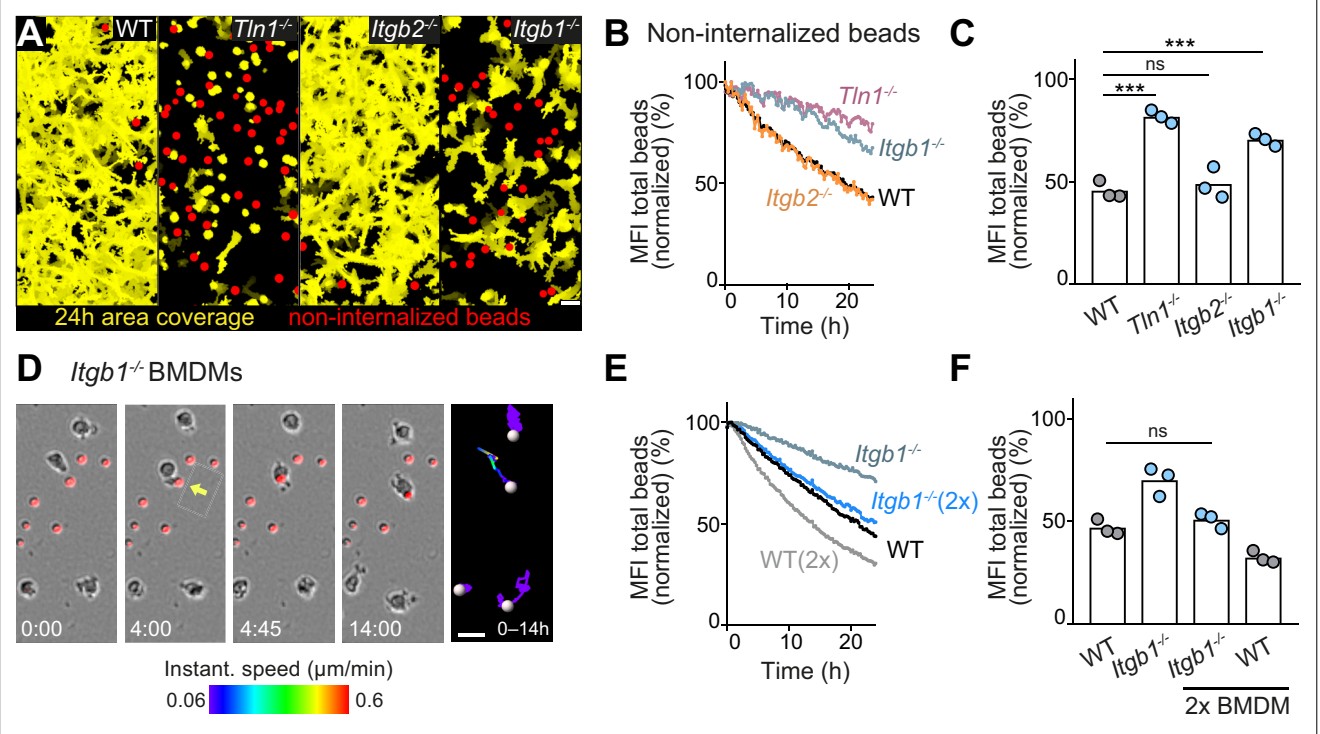

**Figure 6.** Haptokinesis is required for optimal bead removal by macrophage networks. (A–C) Analysis of bone marrow-derived macrophage (BMDM) network sampling activity upon genetic interference with integrin functionality was performed as described in *Figure 5A–E*. (D) Time sequence of an individual *Itgb1⁻/⁻* macrophage is shown, correlating bead sampling and migratory activity as described in *Figure 5G and H*. (E, F) Analysis of *Itgb1⁻/⁻* BMDM network sampling activity upon doubling (2×) the cell number in the BMDM network. Analysis of network sampling as described in *Figure 5C–E*. All bar graphs display the mean; ***p ≤ 0.001, ns: non-significant; Dunnett's multiple comparison (post hoc analysis of variance [ANOVA]). Scale bars: 40 μm (A), 20 μm (D).

The online version of this article includes the following video, source data, and figure supplement(s) for figure 6:

**Source data 1.** Numerical data for the graph in *Figure 6B*.

**Source data 2.** Numerical data for the graph in *Figure 6C*.

**Source data 3.** Numerical data for the graph in *Figure 6E*.

**Source data 4.** Numerical data for the graph in *Figure 6F*.

**Figure supplement 1.** Macrophages require β1 integrins for migration-dependent bead removal in three-dimensional (3D) matrigel.

**Figure supplement 2.** Macrophages do not require β1 integrins bead phagocytosis in suspension.

**Figure supplement 2—source data 1.** Numerical data for the graph in *Figure 6—figure supplement 2*.

**Figure 6—video 1.** Haptokinesis is required for optimal bead removal by macrophage networks.

https://elifesciences.org/articles/75354/figures#fig6video1

individual macrophages that sequentially ingested 8–14 cells over 24–30 hr (*Figure 7—figure supplement 2* and ). Testing the efferocytic capacity of integrin-deficient BMDM networks, we made again the striking observation that β1 integrin-dependent haptokinesis was crucial for optimal surveillance. As observed for the sampling of PS-attached beads that do not release 'find-me' signals, networks of *Tln1⁻/⁻* and *Itgb1⁻/⁻* BMDMs were significantly impaired in the removal of dead neutrophils. The phenotype of *Tln1⁻/⁻* BMDMs was even more pronounced in comparison to *Itgb1⁻/⁻* BMDMs, probably due to migration-independent effects of talin on αv integrins, which contribute to dead cell recognition and uptake (*Lemke, 2019*). In contrast, *Itgb2⁻/⁻* BMDMs showed similar efferocytic activities as WT cells (*Figure 7C and D*; *Figure 7—figure supplement 1*). Haptokinetic movement of mesenchymal-shaped WT BMDMs at speeds of ~0.6 μm/min supported efficient sampling of corpses and surveillance of large areas (*Figure 7E* and ), whereas impaired haptokinesis of *Itgb1⁻/⁻* BMDMs restricted surveillance to smaller regions (*Figure 7F* and *Figure 7—video 3*). Although individual *Itgb1⁻/⁻* macrophages could ingest several dead neutrophils, the slow amoeboid-like movement

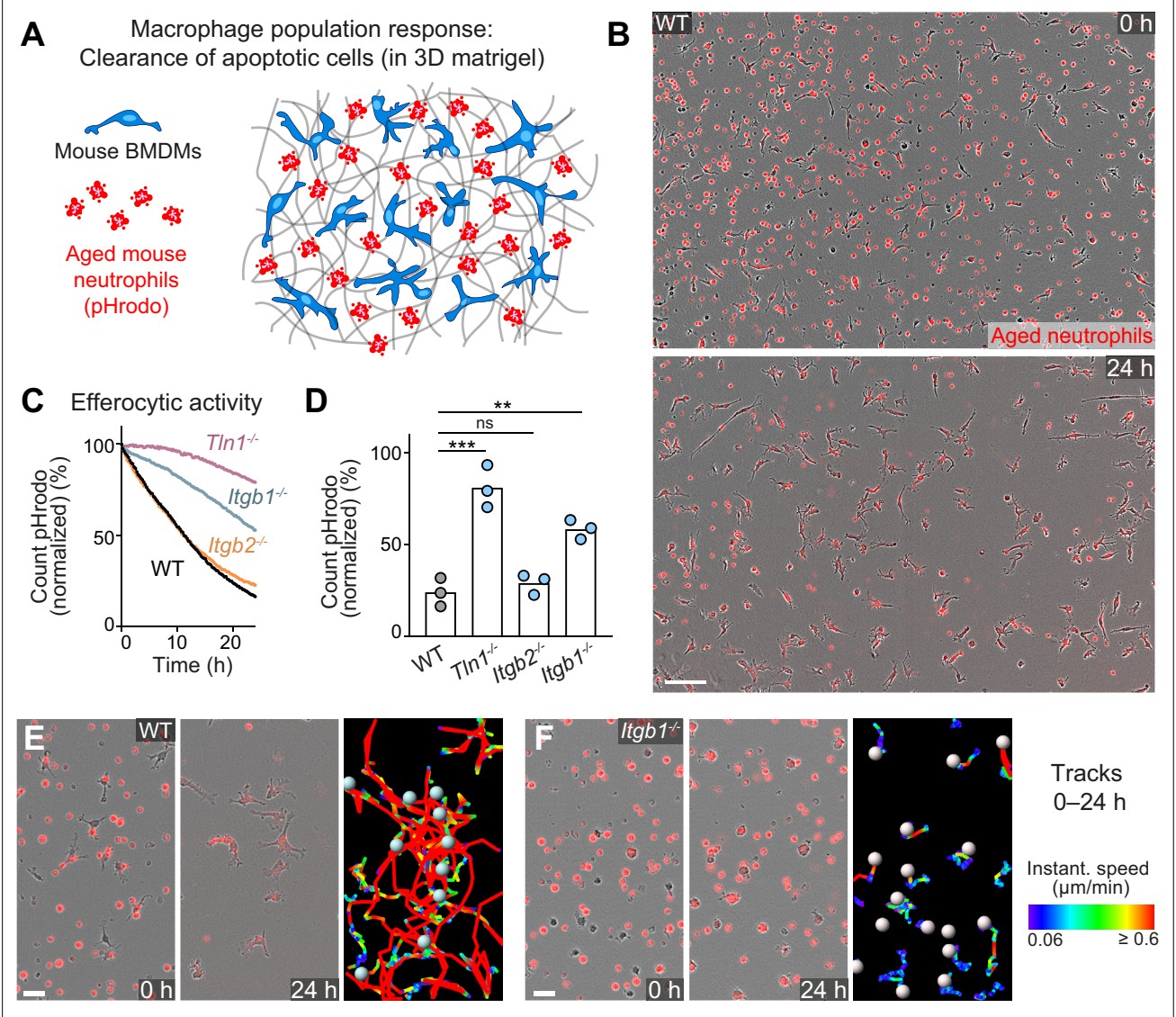

**Figure 7.** Haptokinesis is required for optimal efferocytosis by macrophage networks. (**A**) Scheme for studying the efferocytic response of macrophage networks in three-dimensional (3D) in vitro matrices. (**B**) Live-cell imaging snapshots showing the start (0 hr) and endpoint (24 hr) of dead cell clearance by a population of WT bone marrow-derived macrophages (BMDMs) (unstained). Extracellular, fluorescently pHrodo-labeled aged neutrophils (red, 0 hr) were engulfed and removed by BMDMs over time. The image shows a quarter of the total imaging field of view. Scale bar: 100 µm. (**C, D**) Analysis of dead neutrophil removal by BMDM networks upon genetic interference with integrin functionality. Neutrophil uptake and digestion by macrophages was measured as an object count decline of pHrodo in 15 min intervals over time, presented as (**C**) time-course analysis from one independent experiment (dots in curves are mean values from N = 2–5 technical replicates) (separate wells of matrigel), and as (**D**) 24 hr-mean-values calculated from three biological replicates (n = 3 per genotype). Bars display the mean; ***$p \leq 0.001$, **$p \leq 0.01$, ns: non-significant; Dunnett's multiple comparison (post hoc analysis of variance [ANOVA]). (**E, F**) Correlation of efferocytic and migratory activity in populations of WT (**E**) and *Itgb1*$^{-/-}$ (**F**) BMDMs. Cell tracks over 24 hr are pseudo-colored for instantaneous speed values. Scale bars: 30 µm.

The online version of this article includes the following video, source data, and figure supplement(s) for figure 7:

**Source data 1.** Numerical data for the graph in *Figure 7C*.

**Source data 2.** Numerical data for the graph in *Figure 7D*.

**Figure supplement 1.** Haptokinetic sampling of dead cells optimizes efferocytosis in macrophage networks.

**Figure supplement 1—source data 1.** Numerical data for the graph in *Figure 7—figure supplement 1B*.

**Figure supplement 2.** Haptokinetic sampling of dead cells optimizes efferocytosis in macrophage networks.

**Figure supplement 3.** Haptokinetic sampling of dead cells optimizes efferocytosis in macrophage networks.

**Figure 7—video 1.** Efficient efferocytosis of aged neutrophils by macrophage networks.

*Figure 7 continued on next page*

limited their efferocytic sampling to only nearby corpses (*Figure 7—figure supplement 3*). Thus, our results show an important role for β1 integrins in controlling macrophage movement and protrusiveness, and further highlight haptokinetic sampling as a crucial process for efficient efferocytosis in macrophage networks.

## β1 integrin-dependent surveillance by cortical macrophage networks in lymph nodes

To show the relevance of our in vitro findings for mammalian tissues, we chose to study macrophages located in the T cell cortex of mouse lymph nodes (*Figure 8A*). These cortical macrophages sit on an ECM-rich reticular fiber scaffold, where they form dense cellular networks (*Bellomo et al., 2018*). Previous work has shown that this tissue-resident macrophage type acts as the only professional phagocyte that continuously clears apoptotic cells in the T cell zone of lymph nodes (*Baratin et al., 2017*). Confocal immunofluorescence analysis revealed dense networks of cortical macrophages with elongated protrusions and multi-branched, mesenchymal-like shapes in the T cell zones of WT mice (*Figure 8B*). In contrast, cortical macrophages in lymph nodes of conditional *Itgb1* knockout mice

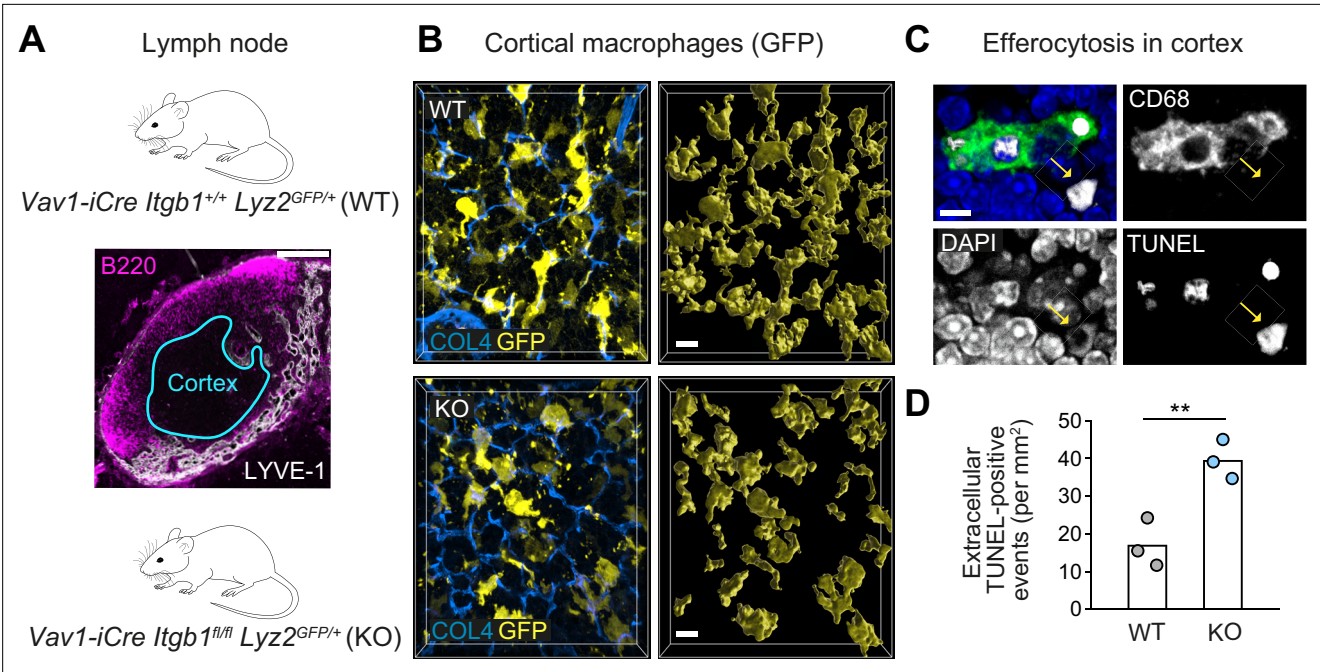

**Figure 8.** β1 integrin-dependent surveillance by cortical macrophage networks in lymph nodes. (**A**) Immunofluorescence staining of a mouse inguinal lymph node. T cell cortex (cyan outline) was defined as B220- and Lyve-1-negative tissue area. (**B**) Confocal immunofluorescence images of GFP-expressing cortical macrophages in WT and conditional *Itgb1*-deficient mice crossed to *Lyz2*$^{GFP/+}$ knock-in mice (left). Collagen IV (COL4) stainings display the cortical reticular fiber network. GFP-based surface representations of macrophage morphologies are shown (right). (**C**) Detection of apoptotic cells by TUNEL method in T cell zones of immuno-stained lymph node sections. TUNEL-positive cells had altered nuclear DAPI stainings, and were found non-internalized (yellow arrow) or internalized by macrophages (stained by CD68). (**D**) Quantification of non-internalized TUNEL-positive cells in T cell cortices. Dots represent individual mice (n = 3 per genotype). Bars display the mean; **p ≤ 0.01, *t* test. Scale bars: 100 μm (**A**), 10 μm (**B**), 5 μm (**C**).

The online version of this article includes the following source data for figure 8:

**Source data 1.** Numerical data for the graph in *Figure 8D*.

were more roundish and showed amoeboid-like morphologies (*Figure 8B*). To evaluate the efferocytic efficiency of cortical macrophage networks, we used the TUNEL method to detect and quantify apoptotic cells in the T cell zones of lymph nodes. TUNEL-positive cells were mostly detected inside cortical macrophages, but were also observed at lower numbers outside the macrophage network (*Figure 8C*). Importantly, mice with *Itgb1*-deficient macrophages showed a significantly increased number of extracellular, non-internalized apoptotic cells in T cell zones in comparison to littermate control mice (*Figure 8D*). Thus, our results confirm the important role of β1 integrin-dependent mesenchymal-like cell shape and motility for efferocytic macrophage networks in vivo.

## Discussion

Given their many important physiological roles, macrophages have evolved mechanisms that ensure robust phago- and efferocytosis. Recent work in *Drosophila* embryos have strengthened this view by showing that hemocytes utilize two distinct modes of engulfment, 'lamellipodial' and 'filopodial' phagocytosis, which provide individual cells phagocytic plasticity to fulfill their clearance functions (*Davidson and Wood, 2020a*). Similar to this previous study, the removal of microbes and dead cells by macrophages has mostly been studied from the viewpoint of a single cell, trying to understand the molecular details of foreign material recognition and uptake (*Elliott and Ravichandran, 2016*; *Mylvaganam et al., 2021*; *Vorselen et al., 2020*). However, mammalian macrophages form cellular networks in almost all tissues, making tissue surveillance a 'group effort' of many individual macrophages. This population behavior is important for the removal of dead cell material in tissues, where the efficiency of detecting, engulfing, and clearing dead cell corpses depends on the efferocytic capacity of a whole macrophage population, which sometimes even requires additional support from non-professional phagocytes (*Damisah et al., 2020*; *Han et al., 2016*). Hence, it is evident that molecules recognizing 'find-me' and 'eat-me' signals and their surface expression levels on individual cells are important determinants for the efferocytic capacity of macrophage populations (*Hughes et al., 2021*; *Rothlin and Ghosh, 2020*).

Here, we investigated how the cell shape and the motility mode of individual macrophages influence the efferocytic capacity of macrophage networks. Our study sought to determine the critical and non-critical parameters that shape the dynamic surveillance behavior of macrophage networks in 3D environments. In particular, we focused on the cytoskeletal control of macrophage motility and migration, the basic processes that enable the formation of directed cell protrusions and displacement and thus realize the interstitial recruitment preceding the engulfment of dead cell corpses. Our findings expand the concept of plasticity to modes of tissue surveillance in macrophage networks. We show that actomyosin contractility is non-critical for sampling macrophage populations, as they can survey their surrounding by two equally efficient strategies: (a) surveillance by migration and (b) surveillance by low-motile protrusion extension. We speculate that both surveillance modes are likely reflected in the heterogeneity of resident macrophages in mammalian tissues (*Blériot et al., 2020*; *Cox et al., 2021*), contributing to the robustness of macrophage network function. Although we know that macrophages of different developmental origin co-exist in many tissues (*Blériot et al., 2020*; *Cox et al., 2021*), we have only limited understanding about their cytoskeletal properties. In future studies it will be interesting to address how distinct macrophage subsets differ in their protrusive and contractile forces and how these factors determine the surveillance potential and adaptation of macrophages to a specific tissue compartment (*Okabe and Medzhitov, 2016*).

As the most important result of our study, we identify that both surveillance strategies critically depend on haptokinesis. Integrin β1-mediated substrate binding controls the 3D mesenchymal-like shape, movement, lamellopodial protrusiveness, and space exploration of individual cells in a sampling macrophage network. Loss of β1 integrin function switches the cells to an amoeboid-like morphology, which does not support random motility and the sampling of particles or dead cells by the macrophage network. In contrast to many other immune cell types (*Lämmermann and Germain, 2014*), we find that randomly migrating mouse macrophages cannot compensate for the loss of integrin function and the associated switch from a mesenchymal- to an amoeboid-like morphology. Previous studies with zebrafish macrophages and hMDMs in 3D matrix gels detected integrin-dependent adhesion structures (*Barros-Becker et al., 2017*; *Van Goethem et al., 2011*), but the functional consequences of integrin depletion for migration remained unexplored. Similar to immature dendritic cells (*Gawden-Bone et al., 2014*), mammalian macrophages form integrin-dependent focal adhesions

and podosomes on 2D surfaces (*Owen et al., 2007*; *Wiesner et al., 2014*). Interestingly, studies with hMDMs invading 3D collagen gels found classical focal adhesion and podosome components (e.g. talin, paxillin, vinculin) at the tip of F-actin-rich cell protrusions together with β1 integrins (*Van Goethem et al., 2011*; *Wiesner et al., 2014*). However, the exact nature of the integrin-dependent adhesion structure promoting macrophage 3D migration remains to be explored in future studies.

To our surprise, we find contrasting integrin demands of random and chemotactic macrophage migration. When talin- or β1 integrin-deficient macrophages were exposed to gradients of high C5a attractant concentrations, these amoeboid-shaped cells polarized their leading edges toward the attractant source and moved chemotactically with speeds similar to control cells. This behavior is reminiscent of integrin-independent 3D chemotaxis of dendritic cells and neutrophils (*Lämmermann et al., 2008*), but at 6- to 16-fold lower average speeds, respectively. Studies in *Drosophila* had shown varying results on integrin dependence for the directed migration of hemocytes toward laser-induced tissue injuries (*Comber et al., 2013*; *Moreira et al., 2013*). Our 2P-IVM experiments on mouse skin confirmed that dermal macrophages do not require β1 integrins during their chemotactic response toward local gradients of wound attractants. Thus, strong chemotactic signals can induce and maintain the polarization of the macrophage actomyosin cytoskeleton, enabling productive migration in the absence of integrin-dependent substrate binding.

Our experiments also touched on the role of Arp2/3 complex-mediated formation of branched actin networks and their functional role for 3D macrophage migration. Arp2/3 activity is crucial for dendritic actin nucleation underneath the plasma membrane, controlling protrusive lamellipodia at the cell edge and thus the mesenchymal phenotype of many cell types (*Rotty et al., 2013*). Moreover, this protein complex is also involved in several other cellular processes, including phagocytosis (*Jaumouillé and Waterman, 2020*; *Papalazarou and Machesky, 2021*). Similar to integrins, we find contrasting demands of Arp2/3 activity for random versus chemotactic 3D migration. Randomly migrating macrophages lost their mesenchymal phenotype upon CK-666 treatment and were impaired in their speed and displacement. Previous studies with macrophages and fibroblasts had already shown that defective Arp2/3 function resulted in loss of lamellipodia and altered the migration phenotype (*Rotty et al., 2017*; *Suraneni et al., 2012*; *Swaney and Li, 2016*; *Wu et al., 2012*). *Arpc2*-deficient BMDMs were found to move with increased speed over ECM-coated 2D surfaces (*Rotty et al., 2017*), in an experimental setup where the promiscuous binding of β2 integrins to cell culture ware influences the macrophage migration phenotype. In contrast, macrophage movement in 3D matrigels strictly depends on β1 integrin-mediated mechanotransduction, causing a substantial migration defect upon Arp2/3 complex inhibition. Yet, movement of CK-666-treated macrophages along a strong chemotactic gradient was unimpaired, which is in agreement with several previous studies on fibroblasts (*Asokan et al., 2014*; *Dimchev et al., 2021*; *Wu et al., 2012*), BMDMs (*Rotty et al., 2017*), and other immune cell types (*Georgantzoglou et al., 2021*; *Leithner et al., 2016*; *Moreau et al., 2015*; *Vargas et al., 2016*), supporting the general notion that dendritic actin networks rather inhibit than support persistent movement along chemotactic cues and in confined environments. As seen for low-adhesive neutrophils and dendritic cells (*Georgantzoglou et al., 2021*; *Leithner et al., 2016*), macrophages also rely on Arp2/3-mediated front actin networks for space exploration in complex environments. However, macrophages strictly depend on integrin-mediated ECM interactions to establish lamellopodial protrusions as exploratory structures. Their directed movement toward strong chemotactic gradients does not require integrins or Arp2/3, which is reminiscent of cell types that perform amoeboid migration, where persistent locomotion depends on actin flow and actomyosin contraction (*Georgantzoglou et al., 2021*; *Leithner et al., 2016*).

When macrophages survey their surrounding for dead cells, they are considered to sense attractants released from dying cells, so called 'find-me' signals, which initiate the formation of directed protrusions and cell displacement toward the dead cell material (*Elliott and Ravichandran, 2016*; *Lemke, 2019*; *Rothlin et al., 2021*). Although we showed that integrin receptors were dispensable for 3D macrophage chemotaxis, integrin-mediated movement and protrusiveness was absolutely critical for efferocytosis by a surveying macrophage network. We speculate that individual dying cells release only low amounts of attractants, which generate chemotactic signals that are too weak and transient to support the full polarization of the macrophage cytoskeleton in the absence of functional integrins. Thus, integrin-dependent stabilization of the mesenchymal-like cell shape is crucial for efficient efferocytosis by macrophage networks. Insufficient haptokinetic sampling compromises the efferocytic

activity of macrophage networks, as shown for β1 integrin-deficient lymph node cortical macrophages whose clearance of apoptotic T cells was impaired.

Our results are particularly relevant for ECM-rich tissues where β1 integrins define the mesenchymal-like shape of endogenous macrophages. Macrophages in the skin dermis and dura mater interact with fibrillar interstitial matrix and basement membranes (BMs), splenic red pulp, and lymph node cortical macrophages locate on top of BM-rich reticular fiber networks, and intravascular Kupffer cells align along the BMs of liver sinusoids. In all these examples, other integrin family members or adhesion receptor systems could not compensate the loss of β1 integrins and restore the mesenchymal-like shape of endogenous macrophages. β2 integrins, which are abundantly expressed on macrophages, appear not to be involved in pro-migratory mechanotransduction in these interstitial environments, and likely serve other macrophage functions (e.g. pattern recognition, complement- and IgG-mediated phagocytosis, cell retention) (*Cui et al., 2018*; *Jaumouillé et al., 2019*; *Torres-Gomez et al., 2020*). αV integrins, which we did not target in this study, may also rather support other macrophage functions (e.g. apoptotic cell uptake, TGFβ activation) (*Kelly et al., 2018*; *Lemke, 2019*). For organs and tissue compartments with largely cellular composition (e.g. brain, glands, epithelial layers), the maintenance of the mesenchymal-like macrophage phenotype may not necessarily require integrin-dependent ECM binding (*Brand et al., 2020*; *Meller et al., 2017*). Instead, haptokinesis for efficient macrophage surveillance might be realized by other adhesion receptor systems, whose identification requires more systematic studies for many of these tissues.

In summary, our study highlights macrophages as a tissue-resident immune cell type that does not follow the general prevailing paradigm of integrin-independent 3D migration of immune cells. Substrate-dependent movement and protrusiveness are critical determinants for space exploration and efficient sampling of macrophage networks. Thus, integrins are not only critical for the migratory processes of intravascular crawling (*Neupane et al., 2020*), extravasation (*Nourshargh et al., 2010*), and invasion (*Arasa et al., 2021*), but also crucial for the interstitial movement of myeloid immune cell subsets. Our mechanistic insights will also likely be relevant for other tissue-resident immune cell types, which have not yet been studied in detail.

# Materials and methods

## Key resources table

| Reagent type (species) or resource | Designation | Source or reference | Identifiers | Additional information |
|---|---|---|---|---|
| Antibody | Anti-CD61 PE (Armenian hamster monoclonal) | BD Biosciences | Cat# 553347; RRID:AB_394800 | FACS(1:400) |
| Antibody | Anti-CD29 PE (Armenian Hamster monoclonal) | BioLegend | Cat# 102208; RRID:AB_312885 | FACS(1:400) |
| Antibody | IgG Isotype Control PE (Armenian hamster monoclonal) | BioLegend | Cat# 400907; RRID:AB_326593 | FACS(1:400) |
| Antibody | Anti-goat Alexa Fluor 488 (Donkey polyclonal) | Thermo Fisher Scientific | Cat# A11055; RRID:AB_2534102 | IF(1:300) |
| Antibody | Anti-rabbit Alexa Fluor 568 (Donkey polyclonal) | Thermo Fisher Scientific | Cat# A10042; RRID:AB_2534017 | IF(1:300) |

*Continued on next page*

*Continued*

| Reagent type (species) or resource | Designation | Source or reference | Identifiers | Additional information |
|---|---|---|---|---|
| Antibody | Anti-GFP Dylight 488 (Goat polyclonal) | Rockland | Cat# 600-141-215; RRID:AB_1961516 | IF(1:750, 1:500) |
| Antibody | Anti-rabbit Alexa Fluor 405 (Goat polyclonal) | Thermo Fisher Scientific | Cat# A-31556; RRID:AB_221605 | IF(1:300, 1:200) |
| Antibody | Anti-rabbit Alexa Fluor 700 (Goat polyclonal) | Thermo Fisher Scientific | Cat# A21038; RRID:AB_10373851 | IF(1:100) |
| Antibody | Anti-collagen IV (Goat polyclonal) | Merck Millipore | Cat# AB769; RRID:AB_92262 | IF(1:200) |
| Antibody | Anti-talin (Mouse monoclonal) | Sigma-Aldrich | Cat# T3287; RRID:AB_477572 | WB(1:1000) |
| Antibody | Anti-rabbit HRP (Pig polyclonal) | Agilent Dako | Cat# P0217; RRID: AB_2728719 | WB(1:5000) |
| Antibody | Anti-actin (Rabbit polyclonal) | Sigma-Aldrich | Cat# A2066; RRID:AB_476693 | WB(1:2000) |
| Antibody | Anti-collagen IV (Rabbit polyclonal) | Abcam | Cat# ab19808; RRID:AB_445160 | IF(1:500) |
| Antibody | Anti-Iba1 (Rabbit polyclonal) | Wako | Cat# 019–19741; RRID:AB_839504 | IF(1:200) |
| Antibody | Anti-LYVE1 (Rabbit polyclonal) | Abcam | Cat# AB14917; RRID:AB_301509 | IF(1:200) |
| Antibody | Anti-mouse HRP (Rabbit polyclonal) | Agilent Dako | Cat# P0161; RRID:AB_2687969 | WB:(1:5000) |
| Antibody | Anti-CD11b PE (Rat monoclonal) | BD Biosciences | Cat# 557397; RRID:AB_396680 | FACS(1:400) |
| Antibody | Anti-CD16/ CD32 Antibody (Rat monoclonal) | BD Biosciences | Cat# 553142; RRID:AB_394657 | FACS(1:250) |
| Antibody | Anti-CD18 PE (Rat monoclonal) | BD Biosciences | Cat# 553293; RRID:AB_394762 | FACS(1:400) |
| Antibody | Anti-CD49d PE (Rat monoclonal) | Thermo Fisher Scientific | Cat# 12-0492-82; RRID:AB_465697 | FACS(1:400) |

*Continued on next page*

*Continued*

| Reagent type (species) or resource | Designation | Source or reference | Identifiers | Additional information |
|---|---|---|---|---|
| Antibody | Anti-CD49e PE (Rat monoclonal) | BD Bioscience | Cat# 557447; RRID:AB_396710 | FACS(1:400) |
| Antibody | Anti-CD49f Alexa Fluor 488 (Rat monoclonal) | BioLegend | Cat# 313608; RRID:AB_493635 | FACS(1:400) |
| Antibody | Anti-CD45R/ B220 Brilliant violet 421 (Rat monoclonal) | BD Bioscience | Cat# 562922; RRID:AB_2737894 | IF(1:200) |
| Antibody | Anti-CD51 PE (Rat monoclonal) | BD Biosciences | Cat# 551187; RRID:AB_394088 | FACS(1:400) |
| Antibody | Anti-CD68 Alexa Flour 488 (Rat monoclonal) | Biolegend | Cat#137011; RRID:AB_2074847 | IF(1:200) |
| Antibody | Anti-CD206 PE-Dazzle (Rat monoclonal) | Biolegend | Cat#141731; RRID:AB_2565931 | IF(1:200) |
| Antibody | Anti-F4/80 Brilliant violet 421 (Rat monoclonal) | Biolegend | Cat# 123137; RRID:AB_2563102 | FACS (1:100) |
| Antibody | Anti-F4/80 PE (Rat monoclonal) | Thermo Fisher Scientific | Cat# MF48004; RRID:AB_10372666 | IF(1:100) |
| Antibody | Anti-F4/80 Alexa Flour 647 (Rat monoclonal) | Thermo Fisher Scientific | Cat# 50-4801-82; RRID:AB_11149361 | FACS(1:100) |
| Antibody | Anti-Integrin β7 Chain PE (Rat monoclonal) | BD Biosciences | Cat# 557498; RRID:AB_396735 | FACS(1:400) |
| Antibody | IgG1 kappa Isotype Control PE (Rat monoclonal) | Thermo Fisher Scientific | Cat# 12-4301-81; RRID:AB_470046 | FACS(1:400) |
| Antibody | IgG2a Isotype Control Alexa Fluor 488 (Rat monoclonal) | Thermo Fisher Scientific | Cat# R2a20; RRID:AB_2556535 | FACS(1:400) |
| Antibody | IgG2a kappa Isotype Control PE (Rat monoclonal) | Thermo Fisher Scientific | Cat# 12-4321-80; RRID:AB_1834380 | FACS(1:400) |

*Continued*

| Reagent type (species) or resource | Designation | Source or reference | Identifiers | Additional information |
|---|---|---|---|---|
| Antibody | IgG2b kappa Isotype Control PE (Rat monoclonal) | Thermo Fisher Scientific | Cat# 12-4031-82; RRID: AB_470042 | FACS(1:400) |
| Antibody | Anti-Ly6G (Rat monoclonal) | Bio X Cell | Cat# BE0075-1; RRID:AB_1107721 | IP(200 µg) |
| Recombinant protein | Annexin V PE | Biolegend | Cat# 640908 | FACS(1:50) |
| Recombinant protein | Recombinant Murine M-CSF | PeproTech | Cat# 315–02 | |
| Chemical, compound, drug | cOmplete, Protease Inhibitor Cocktail (PIC) | Sigma-Aldrich | Cat# 11697498001 | |
| Chemical, compound, drug | Fluoromount G Mounting Medium | SouthernBiotech | Cat# 0100–01 | |
| Chemical, compound, drug | Matrigel Basement Membrane Matrix | Corning | Cat# 354234 | |
| Chemical, compound, drug | Matrigel Growth Factor Reduced (GFR) Basement Membrane Matrix, Phenol Red-free | Corning | Cat# 356231 | |
| Chemical, compound, drug | Cytochalasin D | Merck | Cat# 250255 | |
| Chemical, compound, drug | Y-27632 | Merck | Cat# 688001 | |
| Chemical, compound, drug | CK-666 | Merck | Cat# 182515 | |
| Chemical, compound, drug | Blebbistatin | Merck | Cat# 203390 | |
| Chemical, compound, drug | pHrodo Red SE | Thermo Fisher Scientific | Cat# P36600 | |
| Chemical, compound, drug | PS Lipid Microparticles | Echelon Bioscience | Cat# P-B1PS | |
| Chemical, compound, drug | Tissue Tek | Sakura | Cat# 4583 | |
| Chemical, compound, drug | DAPI | Sigma-Aldrich | Cat# D9542 | |

*Continued on next page*

*Continued*

| Reagent type (species) or resource | Designation | Source or reference | Identifiers | Additional information |
|---|---|---|---|---|
| Commercial assay, kit | Click-iT Plus TUNEL assay | Thermo Fisher Scientific | Cat# C10618 | |
| Commercial assay, kit | Dead Cell Removal Kit | Miltenyi Biotec | Cat# 130-090-101 | |
| Commercial assay, kit | Neutrophil Isolation Kit | Miltenyi Biotec | Cat# 130-097-658 | |
| Commercial assay, kit | Clarity Western ECL Substrate | Bio-Rad | Cat# 170–5060 | |
| Experimental model, mouse strain | *Itgb1^{fl/fl}* | *Potocnik et al., 2000*, provided by R Fässler (MPI of Biochemistry, Martinsried) | MGI:1926498 (*Itgb1^{tm1Ref}*) | |
| Experimental model, mouse strain | *Tln1^{fl/fl}* | *Petrich et al., 2007*, provided by S Monkley and D Critchley (University of Leicester) | MGI:3770513 (*Tln1^{tm4.1Crit}*) | |
| Experimental model, mouse strain | *Itgb2^{−/−}* | *Scharffetter-Kochanek et al., 1998*, provided by M Sixt (IST, Klosterneuburg) | MGI: 1861705 JAX: 003329 (*Itgb2^{tm2Bay}*) | |
| Experimental model, mouse strain | *Vav1-iCre* | *de Boer et al., 2003*, Jackson Laboratory | MGI: 2449949 JAX: 008610 (*Commd10^{Tg(Vav1-iCre)A2Kio}*) | |
| Experimental model, mouse strain | *Cx3cr1^{CRE}* | *Yona et al., 2013*, Jackson Laboratory | MGI: 5467983 JAX: 025524 (*Cx3cr1^{tm1.1(cre)Jung}*) | |
| Experimental model, mouse strain | *Lifeact-GFP* | *Riedl et al., 2010*, provided by R Wedlich-Söldner (University of Münster) | MGI: 4831036 (*Tg(CAG-EGFP)#Rows*) | |
| Experimental model, mouse strain | *Lyz2^{GFP}* | *Faust et al., 2000*, provided by T Graf (CRG, Barcelona) | MGI: 2654931 (*Lyz2^{tm1.1Graf}*) | |
| Experimental model, mouse strain | *Vav1-iCre^{+/-} Itgb1^{fl/fl}* | In-house breeding (this paper) | | |
| Experimental model, mouse strain | *Vav1-iCre^{+/-} Itgb1^{fl/fl} Lyz2^{GFP/+}* | In-house breeding (this paper) | | |
| Experimental model, mouse strain | *Cx3cr1^{CRE/+} Tln1^{fl/fl}* | In-house breeding (this paper) | | |
| Experimental model, mouse strain | *Cx3cr1^{CRE/+} Tln1^{fl/fl} Lifeact-GFP^{+/−}* | In-house breeding (this paper) | | |
| Software, algorithm | Adobe Illustrator 2019 | Adobe | https://www.adobe.com | |

*Continued on next page*

*Continued*

| Reagent type (species) or resource | Designation | Source or reference | Identifiers | Additional information |
|---|---|---|---|---|
| Software, algorithm | BD FACSDiva v6 | BD | https://www.bdbioscibdbio.com/en-us/instruments/researes-instruments/research-software/flow-cytometry-acquisition/facsdiva-software | |
| Software, algorithm | Fiji, ImageJ2 | *Rueden et al., 2017*; *Schindelin et al., 2012* | https://imagej.net | |
| Software, algorithm | FlowJo v10.8.0 | BD | https://www.flowjo.com | |
| Software, algorithm | GraphPad Prism Version 8 | GraphPad | https://www.graphpad.com | |
| Software, algorithm | Image Lab Software | Bio-Rad | https://www.bio-rad.com/de-at/product/image-lab-software | |
| Software, algorithm | Imaris v9.5–9.7 | Bitplane | https://imaris.oxinst.com/versions/9-5 | |
| Software, algorithm | ZEN 2012 SP5 FP3 (black) | ZEISS Microscopy | https://www.zeiss.com/microscopy/int/products/microscope-software/zen.html#modules | |
| Software, algorithm | Incucyte Base Software | Incucyte | https://www.sartorius.com/en/products/live-cell-imaging-analysis/live-cell-analysis-software/incucyte-base-software | |
| Software, algorithm | Incucyte Spheroid Analysis Software Module | Incucyte | https://www.sartorius.com/en/products/live-cell-imaging-analysis/live-cell-analysis-software/incucyte-base-software | |
| Software, algorithm | Ibidi chemotaxis and migration tool | Ibidi | https://ibidi.com/chemotaxis-analysis/171-chemotaxis-and-migration-tool.html?gclid=EAIaIQobChMI2c2LtcbW8wIVgrh3Ch2B8wV9EAAYASAAEgLMbvD_BwE | |

## Experimental model

### Mouse models

All used mouse strains and crosses were on a C57BL/6J background and are listed in the Key resources table. $Itgb1^{fl/fl}$ (*Potocnik et al., 2000*), $Tln1^{fl/fl}$ (*Petrich et al., 2007*), $Itgb2^{-/-}$ (*Scharffetter-Kochanek et al., 1998*), $Commd10^{Tg(Vav1-icre)}$ (*de Boer et al., 2003*), $Cx3cr1^{CRE}$ (*Yona et al., 2013*), $Tg(Lifeact-GFP)$ (*Riedl et al., 2010*), and $Lyz2^{GFP}$ (*Faust et al., 2000*) mouse strains have been described elsewhere. Mice were maintained in a conventional animal facility at the Max Planck Institute of Immunobiology and Epigenetics according to local regulations. Animal breeding and husbandry were performed in accordance with the guidelines provided by the Federation of European Laboratory Animal Science Association and by German authorities and the Regional Council of Freiburg. All mouse strains in this study were without health burden. Mouse strains without fluorescent reporter lines and mouse crosses with $Tg(Lifeact-GFP)$ were only used for organ removal after euthanasia by carbon dioxide exposure. $Vav1$-$iCre$ $Itgb1^{fl/fl}$ $Lyz2^{GFP/+}$ and WT $Lyz2^{GFP/+}$ control mice were used for 2P-IVM. Adult mice (>8 weeks) were age- and sex-matched in all experiments, and littermate animals were used as controls in most experiments. For Cre-expressing mouse strains, Cre-expressing littermate control animals were preferred. A contribution of Cre expression to biological phenotypes was never observed and ruled out through control experiments. Intravital imaging experiments were performed according to study protocols approved by the German authorities and the Regional Council of Freiburg (35–9185.81/G-18/111).

## Method details

### Ear skin and dura mater whole mount staining

Whole mounts of murine ear skin were prepared by splitting the ear in half and by separating the dermal tissue from the cartilage. Split ears were fixed in 1% PFA in PBS for 16 hr at 4°C. The tissue was then blocked and permeabilized by incubating it in wash/staining solution (0.2% Triton X-100, 1% bovine serum albumin [BSA; Sigma-Aldrich] in PBS) for 16 hr on a plate shaker. Primary antibody

staining was performed for 16 hr shaking at 4°C. Subsequently, the tissue was washed three times for 15 min at room temperature in wash/staining solution. Samples were stained with secondary antibody solution for 16 hr shaking at 4°C, followed by an additional three washing steps. To isolate dura mater whole mounts, the cranium skull together with the dura mater were dissected and placed in 4% PFA for 4 hr at 4°C. The dura mater was then peeled away from the cranium skull bones and stained in the same manner as ear skin whole mounts. Antibodies used for labeling ear skin whole mounts were: anti-collagen IV (1:500, Abcam), anti-rabbit Alexa Fluor 405 (1:300, Thermo Fisher Scientific), and anti-CD206 (1:200, Biolegend). Antibodies used for labeling dura maters were: anti-Iba1 (1:200, Wako), anti-collagen IV (1:200, Merck Millipore), anti-goat Alexa Fluor 488 (1:300, Thermo Fisher Scientific) and anti-rabbit Alexa Fluor 568 (1:300, Thermo Fisher Scientific). The tissues were mounted on Superfrost glass slides (Thermo Fisher Scientific) with a coverglass and Fluoromount-G (SouthernBiotech). Image acquisition was performed using an LSM 780 microscope (Zeiss) with a Plan-Apochromat 20× M27 objective (Zeiss) as well as with a Plan-Apochromat 40×/1.4 Oil DIC M27 objective.

## Mouse culture

Mouse BMDMs were generated from bone marrow precursors by standard M-CSF culture. Os coxae, tibia, and femora were dissected from mice and the bone marrow flushed with RPMI. The resulting bone marrow suspension was passed through a 70 µm filter and pelleted in a centrifuge at 330× $g$ for 5 min. Upon re-suspension cells were counted and re-suspended at $5 \times 10^6$ cells/ml in heat-inactivated fetal calf serum (FCS) with 10% DMSO and stored at −80°C until usage. For macrophage differentiation, frozen bone marrow cells were defrosted, washed once with 20 ml RPMI (37°C) at 330× $g$ for 5 min and re-suspended in 10 ml of macrophage medium (RPMI, 10% FCS, 1% penicillin/streptomycin and 20 ng/ml M-CSF). The cell suspension was plated on a 10 cm Petri dish and incubated at 37°C, 5% $CO_2$ (day 0). On days 3 and 5 of the culture, 5 ml of fresh macrophage medium was added on top of the pre-existing medium. Cells were harvested with 20 mM EDTA at day 6 of differentiation. Dead cells were removed using a dead cell removal kit (Miltenyi Biotec) prior to any experiment, in accordance with the manufacturer's instructions.

## Flow cytometry

BMDMs were harvested as described before and Fc receptors were blocked with an anti-mouse CD16/CD32 antibody (1:250, BD Biosciences) in FACS buffer (5% heat-inactivated FCS, 2 mM EDTA in PBS) for 10 min at room temperature. Cells were stained with the desired antibody cocktail for 30 min on ice, followed by three wash steps with FACS buffer (5 min at 300× $g$). Cells were re-suspended in DAPI solution (0.5 µg DAPI in FACS buffer) and incubated for 10 min at room temperature. The cells were then analyzed using an LSR III or LSRFortessa (BD Biosciences) flow cytometer. Data were processed with the FlowJo software (BD Bioscience), where the integrin expression of living (DAPI negative) F4/80-expressing cells (1:100, Invitrogen) was analyzed. Antibodies used were: PE-conjugated anti-CD29 (1:400, Biolegend), PE-conjugate anti-CD11b (1:400, BD Bioscience), Alexa Fluor 488-conjugated anti-CD49f (1:400, Biolegend), rat IgG2a kappa isotype control (1:400, Thermo Scientific), Armenian hamster IgG isotype control (1:400, Biolegend), rat IgG1 kappa isotype control (1:400, Thermo Fisher Scientific), anti-integrin β7 chain (1:400, BD Bioscience), rat IgG2a isotype control (1:400, Thermo Fisher Scientific), rat IgG2b kappa isotype control (1:400, Thermo Fisher Scientific), anti-CD18 (1:400, BD Bioscience), anti-CD49d (1:400, Thermo Scientific), anti-CD51 (1:400, BD Bioscience), anti-CD61 (1:400, BD Bioscience), anti-CD49e (1:400, BD Bioscience), and rat IgG1 kappa isotype control (1:400, Thermo Fisher Scientific). To assess phagocytosis of macrophages in suspension, a 2:1 ratio of fluorescent PS-attached beads and BMDMs suspended in macrophage medium were incubated for 2 hr on a shaker at 37°C and 700 rpm. Afterward cells were Fc blocked in annexin-binding buffer (135 mM NaCl, 5 mM KCl, 5.6 mM glucose, 1.8 mM $CaCl_2$, 1 mM $MgCl_2$, and 20 mM HEPES, pH 7.3) using CD16/32 blocking antibodies (1:250, BD Biosciences), which was followed by labeling in annexin-binding buffer with anti-F4/80 antibodies (1:100, Invitrogen) and annexin V (1:50, Biolegend) for 25 min at 4°C. Cells were then washed two times and re-suspended in annexin-binding buffer containing 0.5 µg/ml DAPI. Results were acquired by flow cytometry using an LSR III (BD Biosciences) and analyzed using FlowJo software (BD Bioscience). Since internalization of PS-attached beads by BMDMs would shield them from annexin V labeling, cells which had acquired a green bead florescence signal, but were still annexin V negative, were defined as having internalized beads.

## Immunoblot analysis

For immunoblot analysis, $5 \times 10^5$ BMDMs were lysed in RIPA buffer (50 mM Tris-HCl, 150 mM NaCl, 0.5% (v:v) NP40, 1% (v:v) Triton X-100, 5 mM EGTA, 5 mM EDTA, 1× cOmplete protease inhibitor cocktail) for 15 min on ice with regular pipetting. Proteins were separated by SDS-PAGE (Bio-Rad) on a 12% polyacrylamide gel, followed by a semi-dry transfer onto a PVDF membrane (Millipore). Nonspecific binding sites were blocked with Tris-buffered saline (TBS) containing 5% BSA and 0.1% (v:v) Tween-20. The membrane was incubated with antibodies against pan-talin (1:1000, Sigma-Aldrich) or actin (1:2000, Sigma-Aldrich) in 0.1% Tween-20% and 5% BSA overnight at 4°C on a shaker. After three washes for 15 min in 0.7% Tween-20 in PBS, the membrane was incubated in secondary antibody solutions (TBS containing 0.1% Tween-20 and 5% BSA, HRP-conjugated secondary antibodies [1:5000, Dako]) at room temperature. Protein bands were visualized with Clarity Western ECL substrate (Bio-Rad), using a ChemiDoc Touch Gel Imaging System (Bio-Rad).

## Mouse neutrophil preparation for efferocytosis assay

Neutrophils were purified from freshly isolated mouse bone marrow cell suspensions using an autoMACS pro-selector cell separator with a MACS neutrophil isolation kit for negative selection in accordance with the manufacturer's instructions (Miltenyi Biotec). Neutrophils were aged overnight in serum-free medium at $3 \times 10^6$ cells/ml at 37°C and 5% $CO_2$. Aged neutrophils were stained prior to use in the 3D efferocytosis assay with 1 ml of 10 µg/ml pHrodo Red SE in HBSS per $2 \times 10^6$ cells for 45 min at 37°C. Cells were then washed twice with RPMI.

## Random migration, efferocytosis, and bead uptake in 3D matrigel

For 3D random migration, efferocytosis, and bead uptake assays, BMDMs were seeded at a concentration of $2.4 \times 10^5$ cells/ml in 40% Matrigel supplemented with 20 ng/ml M-CSF in a 96-well Incucyte image lock plate on ice under sterile conditions. When required, inhibitors were added to the final Matrigel at given concentrations (30 µM Y27632, 2 µM cytochalasin D, 100 µM CK-666 or 60 µM blebbistatin). For 3D efferocytosis and bead uptake assays, aged fluorescent neutrophils or 3 µm sized PS-attached fluorescent beads (Echelon Biosciences) were added to the final Matrigel cell suspension at $4 \times 10^5$ neutrophils or $2–2.4 \times 10^5$ beads per ml, respectively. After adding Matrigel solution to wells of the ice-cold 96-well Incucyte image lock plates, the plates were centrifuged for 3 min at 75× $g$ and 4°C to bring the macrophages into the same focal plane for imaging. This step was required because the Incucyte S3 live-cell analysis system operates with an autofocus mode. Plates were then transferred to a 37°C cell incubator to initiate the fast polymerization process of the Matrigel matrix and kept for 30 min to ensure complete polymerization of the gel. In this experimental setup the majority of macrophages moves over 24 hr primarily in the lower part of the gel, but cells also move vertically in the 3D gel and leave the autofocus plane. Samples were then left at room temperature for 10 min before 200 µl of macrophage medium, with or without the addition of inhibitors, were added on top of the Matrigel. Assays were acquired using an Incucyte S3 live-cell analysis system (Sartorius). Each well was imaged in 15 min intervals for 24–30 hr with the image lock module and the 20× objective. Fluorescent signals were acquired with the inbuilt dual color module 4614, with the visualization of PS-attached fluorescent beads in the green imaging channel or pHrodo Red SE labeled neutrophils in the red imaging channel. For cell morphology visualization during random migration, Lifeact-GFP expressing BMDMs were additionally acquired by fluorescent confocal microscopy after 24 hr in the gel. Here, $5 \times 10^5$ macrophages/ml in 20 ng/ml M-CSF-supplemented 40% Matrigel were seeded in µ angiogenesis slides (Ibidi). Image acquisition was performed on an LSM 780 microscope (Zeiss), fitted with a Plan-Apochromat 40×/1.4 Oil DIC M27 objective.

## Macrophage chemotaxis in 3D matrigel

Ibidi µ chemotaxis slides were used for 3D chemotaxis assays. Ten µl of Matrigel macrophage suspensions (40% phenol red free growth factor reduced Matrigel, macrophages at $3 \times 10^6$ cells/ml and 20 ng/ml M-CSF) with or without inhibitors (30 µM Y27632, 2 µM cytochalasin D or 100 µM CK-666) were added to each center port of the chemotaxis slide. The slide was then left to rest at room temperature for 7 min, followed by an incubation of 7 min at 37°C and 5% $CO_2$. The slide was finally left to settle for 5 min at room temperature. Peripheral ports on each side of the Matrigel were filled with 65 µl macrophage medium with or without the before-mentioned inhibitors. The slides were

incubated for 1 hr at 37°C and 5% $CO_2$. A C5a gradient was subsequently generated by adding 15 µl macrophage medium (with or without inhibitors) containing 60 nM C5a to both ports on one side of the Matrigel. On the opposite side, 15 µl macrophage medium were added to both ports (with or without inhibitors). The slide was then loaded into the Incucyte S3 live-cell analysis system (Sartorius) using a custom-made slide mount and cells were imaged using the spheroid module in 15 min intervals for 24–30 hr with the 20× objective.

## Tissue processing and immunofluorescence staining

Organs (spleen, liver, lymph nodes) were harvested and placed in 1% PFA at 4°C overnight. After incubation in a 30% sucrose solution for 8 hr, organs were embedded in molds with Tissue Tek and stored at −20°C. A Leica CM3050 S Cryostat was used to cut tissue into 20 µm thin tissue sections, which were mounted on Superfrost glass slides and stored at −20°C until further processing. For immunofluorescence stainings, samples were blocked in blocking/staining solution (0.1% Triton X-100, 1% BSA in PBS) for 2 hr at room temperature. The blocking buffer was removed and the tissue was stained overnight with primary antibodies in staining solution in a humidified chamber at 4°C. Slides were washed three times with PBS before being incubated for 4 hr in a humidified chamber with the secondary antibodies in staining solution. After staining, the slides were washed a further three times in PBS and mounted with a coverglass using Fluoromount-G (SouthernBiotech). Liver and spleen sections were stained with anti-collagen IV (1:500, Abcam), anti-rabbit Alexa Fluor 405 (1:200, Thermo Fisher Scientific) and PE-conjugated anti-F4/80 (1:100, Thermo Fisher Scientific) antibodies. In addition to staining with anti-collagen IV (1:500, Abcam) and anti-rabbit Alexa Fluor 405 (1:200, Thermo Fisher Scientific) antibodies, the endogenous GFP expression of lymph node sections from *Lyz2*<sup>GFP/+</sup> containing mice were amplified using an anti-GFP Dylight 488 antibody (1:750, Rockland). TUNEL stainings of lymph node sections were carried out using the Click-iT Plus TUNEL assay kit (Invitrogen). Tissue sections were treated with 2% $H_2O_2$ in methanol for 20 min at room temperature, followed by two washes in PBS. Samples were permeabilized with 0.01% Triton X-100, 0.1% sodium citrate in deionized water. The tissue was then rinsed with deionized water and incubated for 10 min with terminal deoxynucleotidyl transferase (TdT) buffer at 37°C. The TdT buffer was replaced with the TdT reaction mix and samples were incubated for 60 min at 37°C. Tissue sections were rinsed again with deionized water and treated with 0.1% Triton X-100, 3% BSA in PBS for 5 min. Subsequently, samples were rinsed once with PBS and incubated with the TUNEL reaction cocktail for 30 min at 37°C. Finally, the tissue sections were washed once with 3% BSA in PBS followed by one rinse with PBS. Antibody staining was then performed on the tissue sections as mentioned above. Antibodies used were: anti-LYVE1 (1:200, Abcam), anti-CD68 (1:200, BioLegend), anti-B220 (1:200 BD Horizon), anti-GFP Dylight 488 (1:500, Rockland), anti-collagen IV (1:500, Abcam), anti-rabbit Alexa Fluor 405 (1:300, Thermo Fisher Scientific), and anti-F4/80 (1:100, Invitrogen). Images were acquired using an LSM 780 microscope (Zeiss) equipped with a Plan-Apochromat 20× M27 objective (Zeiss) or a Plan-Apochromat 40×/1.4 Oil DIC M27 objective.

## Imaging analysis

Tracking analysis was performed with the manual tracking function of Imaris 9.5.1–9.7.1 (Bitplane). For the random 3D migration assays, viable cells in a randomly chosen region were manually tracked on a frame-by-frame basis. Cells that underwent cell division during the imaging period were excluded. For the 3D chemotaxis assays, BMDMs on the side of the gel facing the C5a gradient were tracked. In rare cases, biological replicates were excluded from analysis when macrophages did not respond to the attractant and only very few control cells performed directed migration. Track visualizations for random migration were generated using the Imaris spot module. Visualizations of macrophage chemotaxis tracks were generated by exporting track coordinates from Imaris and by importing them into the Ibidi chemotaxis and migration tool software. Static tissue images were visualized using the Imaris volume and surface features. Cell circularity was manually measured using ImageJ/Fiji and the freehand selection tool, drawing the outline of the cell for every frame. Macrophage scanning and area coverage was visualized by creating binary masks for each frame of the phase-contrast channel using the ImageJ MorphoLibJ plugin. Binary images of all timepoints were combined using the Time-Lapse Color Coder plugin. The time projection image was then merged with the phase-contrast and green fluorescence (beads) channel. The spot function of Imaris was used to generate spots for all

green events. All spots that resided within macrophages (as defined by the phase-contrast channel) were manually removed, so that only extracellular (non-cleared beads) were visualized. For the 2P-IVM analysis of the dermal macrophage chemotactic response, cell bodies of dermal macrophages were manually tracked over 90 min. Cell protrusions were tracked from the onset of protrusion formation until protrusions reached their maximum extension. All protrusions from all biological replicates were tracked and the results statistically analyzed. Bead uptake and neutrophil efferocytosis analysis was performed with the in-built analysis software of the Incucyte S3 instrument. Optimal parameters were manually defined for an analysis batch. Following this, a mask of the fluorescent signal was generated for all wells and timepoints. Within these masks, object counts and florescence intensity were analyzed. For the neutrophil efferocytosis assays, the Incucyte analysis results were further validated using an Imaris spot function analysis. Here, start and endpoint neutrophil florescence signals were used to generate spots for each neutrophil, which were then manually assigned as being either inside or outside of a macrophage. Using these designations the percentage removal of neutrophils was calculated. Cell protrusiveness was analyzed using a modified timeseries-based Sholl analysis and performed by tracking 10 random cells per condition. This entailed overlaying each cells center points with a bull's-eye containing concentric rings (Sholl shells) at 25 μm intervals. The number of occasions a Sholl shell was intersected by a cellular process was counted for all 15 min time intervals in a 24 hr timeframe. This type of analysis was able to capture both multiple branch-based and elongation-based protrusiveness (*Figure 5—figure supplement 2*). The TUNEL assay analysis was performed with ImageJ. T cell zones (B220- and LYVE1-negative regions) were marked as regions of interest (ROI). The number of TUNEL-positive events in the ROI, which did not overlap with a CD68$^+$ macrophage stain, was quantified using the multi-point counter on a *z*-projection.

## 2P-IVM of macrophage chemotaxis

2P-IVM of directed macrophage migration toward a sterile laser-induced wound injury was performed in the absence of neutrophils. To avoid any contribution of neutrophils to this response, neutrophils were depleted by one intraperitoneal injection with 200 μg of anti-Ly6G antibody diluted in PBS the day before imaging. Macrophage populations were visualized by GFP fluorescence in *Vav1-iCre Itgb1$^{fl/fl}$ Lyz2$^{GFP/+}$* and WT *Lyz2$^{GFP/+}$* mice. For the experiment mice were anesthetized using isoflurane (cp-pharma; the isoflurane was vaporized in an oxygen-air mixture; 2% isoflurane was used for induction and 1–1.5% was used for maintenance). The anesthetized mouse was placed in a lateral recumbent position on a custom-made imaging platform, so that the ventral side of the ear pinna rested on a coverslip. The ear was immobilized with a strip of Hansaplast tape, which was lightly stretched over the ear and the imaging platform. 2P-IVM was performed using an LSM 780 NLO microscope (Zeiss) enclosed in a custom-built environmental chamber that was maintained at 32°C using heated air (*Kienle et al., 2021*). Anesthetized mice were kept in the heated environmental chamber for 15–30 min until the ear tissue had settled. Once the tissue was stable, a focal skin injury was induced by a focused 2P laser pulse at an approximate laser intensity of 80 mW. A circular ROI of 15–30 μm in diameter was defined in one focal plane of the collagenous ear dermis, followed by laser scanning at 920 nm wavelength until tissue coagulation started within 1–3 s. Image acquisition was started immediately after laser-induced tissue damage. A water immersion C-Apochromat 40×/1.2 with corrector M27 objective was used for image acquisition. The microscope system was fitted with four external non-descanned photomultiplier tube detectors in the reflected light path. Fluorescence excitation was provided by an Insight Ds + (Spectra Physics) tuned to 920 nm for GFP excitation and the generation of collagen second harmonic signal. Non-descanned detectors collected the emitted light. Images were mainly captured toward the anterior half of the ear pinna where hair follicles are sparse. For 4D data sets, 3D stacks were captured every 1 min. Raw imaging data were processed with Imaris software version 9.1.2 (Bitplane). All movies are displayed as 2D maximum-intensity projections of 10–15 μm thick *z*-stacks. As the laser ablation turns the circular injured tissue autofluorescent in several channels, we masked the GFP autofluorescence of this region with Imaris-based image processing for data presentation.

## Statistical analyses

Sample size was determined prior to experiment for all experiments used for hypothesis testing (i.e. data that include statistical inference). Technical replicates of one biological replicate were designated

with 'N', biological replicates were designated with 'n'. Sample sizes for technical replicates (i.e. the tracking of randomly chosen migrating cells) in one biological replicate were considered based on the mean and standard deviation of WT macrophage speed during migration. We defined a 30% reduction of mean speed at a power of 0.95 as biologically meaningful effect, determining a sample size of N = 25. Reproducibility of the experimental findings was verified using biological replicates, which were performed as independent experiments. Experimental groups were defined by inhibitor treatment or by the genotype. Sample sizes for biological replicates in cell culture experiments (i.e. BMDM cultures generated from different individual mice) aimed for a minimum of mouse donors to reduce the number of laboratory animals. Sample size for animal experimentation was determined according to animal welfare guidelines. Blinding was not relevant for experiments with genotyping groups because all experimental groups were treated the same. Unpaired two-tailed *t* tests and analysis of variance (ANOVA) were performed after data were confirmed to fulfill the criteria of normal distribution, otherwise two-tailed Mann-Whitney *U* tests or Kruskal-Wallis tests were applied. The D'Agostino & Pearson normality test was performed for group sizes over 10, for group sizes under 10 the Shapiro-Wilk normality test was performed. If overall ANOVA or Kruskal-Wallis tests were significant, we performed post hoc test with pair-wise comparisons (ANOVA: Dunnett, Kruskal-Wallis: Dunn). Analyses were performed with GraphPad Prism-software (version 8.3.1). Asterisks indicate significance (*p ≤ 0.05, **p ≤ 0.01, ***p ≤ 0.001). NS indicates non-significant difference (p > 0.05). For further statistical details, see *Supplementary file 1*.

## Acknowledgements

We thank K Ganter for technical assistance and A Rambold, J Zimmermann, and K Glaser for critically reading the manuscript. We thank R Fässler, C Brakebusch, R Wedlich-Söldner, S Monkley, D Critchley, M Sixt, and T Graf for providing mouse lines for this study. This work was funded by the Deutsche Forschungsgemeinschaft (DFG, German Research Foundation), Project-IDs 259373024 (CRC/TRR 167) and 89986987 (SFB 850), and by the Max Planck Society.

## Additional information

### Funding

| Funder | Grant reference number | Author |
|---|---|---|
| Max Planck Society | | Tim Lämmermann |
| Deutsche Forschungsgemeinschaft | 259373024 | Tim Lämmermann |
| Deutsche Forschungsgemeinschaft | 89986987 | Tim Lämmermann |

The funders had no role in study design, data collection and interpretation, or the decision to submit the work for publication.

### Author contributions

Neil Paterson, Conceptualization, Data curation, Formal analysis, Investigation, Methodology, Validation, Visualization, Writing – review and editing; Tim Lämmermann, Conceptualization, Data curation, Formal analysis, Funding acquisition, Investigation, Methodology, Project administration, Supervision, Validation, Visualization, Writing – original draft, Writing – review and editing

### Author ORCIDs

Neil Paterson ⓘ http://orcid.org/0000-0001-9563-5874
Tim Lämmermann ⓘ http://orcid.org/0000-0002-8553-118X

### Ethics

Mice were maintained in a conventional animal facility at the Max Planck Institute of Immunobiology and Epigenetics, Freiburg, according to local regulations. Animal breeding and husbandry were performed in accordance with the guidelines provided by the Federation of European Laboratory

Animal Science Association and by German authorities and the Regional Council of Freiburg (35-918564/1.1). Research was conducted in accordance with the TierSchG (German animal protection law) and the TierSchVersV (German regulation on the protection of animal in scientific experiments). Intravital imaging experiments were performed according to animal study protocols that have been ethically assessed and approved by the German authorities and the Regional Council of Freiburg (35-9185.81/G-18/111).

## Decision letter and Author response
Decision letter https://doi.org/10.7554/eLife.75354.sa1
Author response https://doi.org/10.7554/eLife.75354.sa2

## Additional files

### Supplementary files
• Transparent reporting form
• Supplementary file 1. Table for overview on statistical tests.

### Data availability
All data generated or analysed during this study are included in the manuscript, supplementary videos and supporting files. Source data files have been provided for Figures 1, 3, 4, 5, 6, 7, 8 and figure supplements. No new unique reagents and no new codes were generated in this study. Previously published datasets were not used in this study.

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
