## [Editor Report]

You have demonstrated that macrophages utilise integrins for tissue surveillance and network formation in a manner distinct from other leukocytes types. You have put this in context with major mechanisms for forming F-actin based protrusions including Arp2/3 dependent branched F-actin. Your work will be of great interest to immunologists, cell biologies and tissue engineers.

---

## [Decision Letter]

**Decision letter after peer review:**

Thank you for submitting your article "Macrophage network dynamics depend on haptokinesis for optimal local surveillance" for consideration by *eLife*. Your article has been reviewed by 3 peer reviewers, including Michael L Dustin as Reviewing Editor and Reviewer #1, and the evaluation has been overseen by Carla Rothlin as the Senior Editor.

Recommendations for the Authors:

The reviewers identify two areas for essential revision that don't require new experiments, but require rewriting/additional explanation and some tempering of conclusions. The reviewer comments are presented below under headings A and B.

A. Role of F-actin based protrusions not fully explored in inhibitory studies- this may be adequately addressed by thorough discussion of work of Bear and colleagues- for example- https://www.ncbi.nlm.nih.gov/pmc/articles/PMC5601320/

1) While actomyosin networks and cell adhesions might be slightly more resistant to the immediate effects of cyto D, they will absolutely be disrupted by this drug. i.e. F-actin is ultimately upstream of actomyosin contractility and talin based adhesion; without any actin polymerisation you can have neither. As such, cyto D is a broad inhibitor of all three of these cytoskeletal processes. Arp2/3 complex inhibition through CK-666 would provide a more targeted means of disrupting dendritic actin and therefore mesenchymal protrusions. CK-666 inhibition should be contrasted to ROCK inhibition throughout this work. This is especially true of the efferocytosis experiments, where Cyto D treatment can only operate as a crude control (phagocytic uptake requires F-actin). A more targeted approach using CK-666 should be taken to truly dissect the distinct surveillance strategies outlined.

2) While ROCK inhibition with Y27632 is an acceptable means of reducing actomyosin contractility, direct inhibition with blebbistatin should be considered, if only in Figure 1.

3) The chemotaxis/phagocytosis data should be put in context of the extensive work from the Bear lab (UNC), any reference to which is surprisingly lacking from the paper.

4) The combined inhibition of both mesenchymal protrusions/dendritic actin (CK-666) and amoeboid protrusive force /actomyosin contractility (Y27632 or blebbistatin) or cell adhesion (talin KO) and amoeboid protrusive force/actomyosin contractility (Y27632 or blebbistatin) would be expected to inhibit all chemotaxis/efferocytosis and should be tested. (Note that the reviewers collectively didn't consider this experiment essential, but that tempering claims and discussing Bear lab work with CK666 could address this).

5) In the discussion the authors state that: macrophages can "survey their surrounding by two equally efficient strategies: (a) migration-based surveillance by contractile, low-protrusive cells, or (b) protrusion-based surveillance by less contractile, low-migrating cells."

The data presented does not support conclusion (a). It can be argued that migration-based surveillance need not be contractile (amoeboid) based, but may also be (and from data appears dominantly to be) driven by highly protrusive and motile, mesenchymal-like motility. In other words, macrophages can survey area through motility (mesenchymal or amoeboid) or non-motile protrusion extension. Again, the use of CK-666 would allow the authors to make a stronger conclusion here.

B) Are matrigel substrates really 3D as cells appear to settle and migrate in 2D. Are these matrix conditions in which dendritic cells or neutrophil would display much faster chemotaxis as implied in discussion of results? Please expand on methods.

---

## [Author Response]

The reviewers identify two areas for essential revision that don't require new experiments, but require rewriting/additional explanation and some tempering of conclusions. The reviewer comments are presented below under headings A and B.A. Role of F-actin based protrusions not fully explored in inhibitory studies- this may be adequately addressed by thorough discussion of work of Bear and colleagues- for example- https://www.ncbi.nlm.nih.gov/pmc/articles/PMC5601320/1) While actomyosin networks and cell adhesions might be slightly more resistant to the immediate effects of cyto D, they will absolutely be disrupted by this drug. i.e. F-actin is ultimately upstream of actomyosin contractility and talin based adhesion; without any actin polymerisation you can have neither. As such, cyto D is a broad inhibitor of all three of these cytoskeletal processes. Arp2/3 complex inhibition through CK-666 would provide a more targeted means of disrupting dendritic actin and therefore mesenchymal protrusions. CK-666 inhibition should be contrasted to ROCK inhibition throughout this work. This is especially true of the efferocytosis experiments, where Cyto D treatment can only operate as a crude control (phagocytic uptake requires F-actin). A more targeted approach using CK-666 should be taken to truly dissect the distinct surveillance strategies outlined.

We completely agree with this comment and now refer to cytochalasin D as an “F-actin disrupting drug” throughout the manuscript. In the course of our study, we had already performed experiments with the suggested Arp2/3 complex inhibitor CK-666, which we are now providing with the manuscript and also address in the discussion of the paper:

1. As expected, inhibition of dendritic actin networks leads to a more roundish, amoeboid-like morphology of macrophages, which only show thin rudimentary cell protrusions (Figure 1, figure supplement 1, C–D, and Video 2). However, the increase in cell circularity is not as pronounced for *Tln1*-deficient bone marrow-derived macrophages (BMDMs). CK-666 induced disruption of dendritic actin networks also leads to impaired random motility in the majority of macrophages (Figure 1, figure supplement 1, E, and Video 2), but is not as pronounced as for integrin-deficient macrophages migrating in 3D matrigels. In our discussion, we compare these findings to published work on the role of Arp2/3 for fibroblasts (Asokan et al., 2014; Dimchev et al., 2021; Suraneni et al., 2015; Suraneni et al., 2012; Wu et al., 2012) and BMDMs migrating on 2D surfaces (Rotty et al., 2017).

2. When investigating the chemotactic movement of CK-666 treated macrophages toward strong soluble gradients of C5a, we did not see any impaired directed migration of WT macrophages and a trend to increased migration speeds (Figure 3, figure supplement 1, and Video 3). This finding is in agreement with several previous studies on fibroblasts (Dimchev et al., 2021; Wu et al., 2012), BMDMs (Rotty/Bear, 2017) and other immune cell types (Georgantzoglou et al., 2021; Leithner et al., 2016; Moreau et al., 2015; Vargas et al., 2016), supporting the general notion that dendritic actin networks rather inhibit than support persistent movement along chemotactic cues and in confined environments.

We had also performed efferocytosis experiments with CK-666, but observed extremely long interaction times between macrophages and dead neutrophils before the final uptake of the dead cell corpse. This indicated to us that CK-666 does not only have effects on migration, but very likely also on the formation of the efferocytic cup in BMDMs. This observation is in line with phagocytosis defects that have been observed for Arp2/3 complex-deficient macrophages (Jaumouille et al., 2019; Jaumouille and Waterman, 2020; May et al., 2000; Rotty et al., 2017; Vorselen et al., 2021). It should be noted that the previously reported “dispensable” role of Arp2/3 for FcR-mediated phagocytosis was only reported for the uptake of 2-µm small beads, but not for bigger particles and other forms of phagocytosis (Rotty et al., 2017). Although we observed decreased efferocytosis in our matrigel assay upon CK-666 treatment, these results remained inconclusive regarding the contribution of impaired migration versus impaired phagocytosis. These observations will require more detailed investigations, which is why we did not include these data. They also go beyond the scope of this study, where we focused on the contribution of migration dynamics for macrophage networks, highlighting the particular role of integrin-mediated adhesions in this process.

References:

Asokan, S. B., Johnson, H. E., Rahman, A., King, S. J., Rotty, J. D., Lebedeva, I. P., Haugh, J. M., and Bear, J. E. (2014, Dec 22). Mesenchymal chemotaxis requires selective inactivation of myosin II at the leading edge via a noncanonical PLCgamma/PKCalpha pathway. Dev Cell, 31(6), 747-760. https://doi.org/10.1016/j.devcel.2014.10.024

Dimchev, V., Lahmann, I., Koestler, S. A., Kage, F., Dimchev, G., Steffen, A., Stradal, T. E. B., Vauti, F., Arnold, H. H., and Rottner, K. (2021). Induced Arp2/3 Complex Depletion Increases FMNL2/3 Formin Expression and Filopodia Formation. Front Cell Dev Biol, 9, 634708. https://doi.org/10.3389/fcell.2021.634708

Georgantzoglou, A., Poplimont, H., Lämmermann, T., and Sarris, M. (2021). Interstitial leukocyte navigation through a search and run response to gradients. bioRxiv, 2021.2003.2003.433706. https://doi.org/10.1101/2021.03.03.433706

Jaumouille, V., Cartagena-Rivera, A. X., and Waterman, C. M. (2019, Nov). Coupling of beta2 integrins to actin by a mechanosensitive molecular clutch drives complement receptor-mediated phagocytosis. Nat Cell Biol, 21(11), 1357-1369. https://doi.org/10.1038/s41556-019-0414-2

Jaumouille, V., and Waterman, C. M. (2020). Physical Constraints and Forces Involved in Phagocytosis. Front Immunol, 11, 1097. https://doi.org/10.3389/fimmu.2020.01097

Leithner, A., Eichner, A., Muller, J., Reversat, A., Brown, M., Schwarz, J., Merrin, J., de Gorter, D. J., Schur, F., Bayerl, J., de Vries, I., Wieser, S., Hauschild, R., Lai, F. P., Moser, M., Kerjaschki, D., Rottner, K., Small, J. V., Stradal, T. E., and Sixt, M. (2016, Nov). Diversified actin protrusions promote environmental exploration but are dispensable for locomotion of leukocytes. Nat Cell Biol, 18(11), 1253-1259. https://doi.org/10.1038/ncb3426

May, R. C., Caron, E., Hall, A., and Machesky, L. M. (2000, Apr). Involvement of the Arp2/3 complex in phagocytosis mediated by FcgammaR or CR3. Nat Cell Biol, 2(4), 246-248. https://doi.org/10.1038/35008673

Moreau, H. D., Lemaitre, F., Garrod, K. R., Garcia, Z., Lennon-Dumenil, A. M., and Bousso, P. (2015, Sep 29). Signal strength regulates antigen-mediated T-cell deceleration by distinct mechanisms to promote local exploration or arrest. Proc Natl Acad Sci U S A, 112(39), 12151-12156. https://doi.org/10.1073/pnas.1506654112

Rotty, J. D., Brighton, H. E., Craig, S. L., Asokan, S. B., Cheng, N., Ting, J. P., and Bear, J. E. (2017, Sep 11). Arp2/3 Complex Is Required for Macrophage Integrin Functions but Is Dispensable for FcR Phagocytosis and in Vivo Motility. Dev Cell, 42(5), 498-513 e496. https://doi.org/10.1016/j.devcel.2017.08.003

Suraneni, P., Fogelson, B., Rubinstein, B., Noguera, P., Volkmann, N., Hanein, D., Mogilner, A., and Li, R. (2015, Mar 1). A mechanism of leading-edge protrusion in the absence of Arp2/3 complex. Mol Biol Cell, 26(5), 901-912. https://doi.org/10.1091/mbc.E14-07-1250

Suraneni, P., Rubinstein, B., Unruh, J. R., Durnin, M., Hanein, D., and Li, R. (2012, Apr 16). The Arp2/3 complex is required for lamellipodia extension and directional fibroblast cell migration. J Cell Biol, 197(2), 239-251. https://doi.org/10.1083/jcb.201112113

Vargas, P., Maiuri, P., Bretou, M., Saez, P. J., Pierobon, P., Maurin, M., Chabaud, M., Lankar, D., Obino, D., Terriac, E., Raab, M., Thiam, H. R., Brocker, T., Kitchen-Goosen, S. M., Alberts, A. S., Sunareni, P., Xia, S., Li, R., Voituriez, R., Piel, M., and Lennon-Dumenil, A. M. (2016, Jan). Innate control of actin nucleation determines two distinct migration behaviours in dendritic cells. Nat Cell Biol, 18(1), 43-53. https://doi.org/10.1038/ncb3284

Vorselen, D., Barger, S. R., Wang, Y., Cai, W., Theriot, J. A., Gauthier, N. C., and Krendel, M. (2021, Oct 28). Phagocytic 'teeth' and myosin-II 'jaw' power target constriction during phagocytosis. eLife, 10. https://doi.org/10.7554/eLife.68627

Wu, C., Asokan, S. B., Berginski, M. E., Haynes, E. M., Sharpless, N. E., Griffith, J. D., Gomez, S. M., and Bear, J. E. (2012, Mar 2). Arp2/3 is critical for lamellipodia and response to extracellular matrix cues but is dispensable for chemotaxis. Cell, 148(5), 973-987. https://doi.org/10.1016/j.cell.2011.12.034

2) While ROCK inhibition with Y27632 is an acceptable means of reducing actomyosin contractility, direct inhibition with blebbistatin should be considered, if only in Figure 1.

We have now also included our experimental data for the effect on blebbistatin treatment on the random motility of WT macrophages (Figure 1, figure supplement 1, A,B). Blebbistatin treatment pheno-copies the effects of Y27632 inhibition on randomly migrating macrophages in matrigel.

3) The chemotaxis/phagocytosis data should be put in context of the extensive work from the Bear lab (UNC), any reference to which is surprisingly lacking from the paper.

Please see point 1. We have included this into our discussion.

4) The combined inhibition of both mesenchymal protrusions/dendritic actin (CK-666) and amoeboid protrusive force /actomyosin contractility (Y27632 or blebbistatin) or cell adhesion (talin KO) and amoeboid protrusive force/actomyosin contractility (Y27632 or blebbistatin) would be expected to inhibit all chemotaxis/efferocytosis and should be tested. (Note that the reviewers collectively didn't consider this experiment essential, but that tempering claims and discussing Bear lab work with CK666 could address this).

We have now also included our experimental data for the effect of Y-27632 treatment on chemotaxing Talin-deficient macrophages (Figure 3, figure supplement 2, C). Similar to WT macrophages, Y27632 treatment impairs the chemotactic migration of *Tln1^−/−^* BMDMs, supporting an important role of actomyosin contractility as amoeboid protrusive force for integrin-independent macrophage migration.

We agree that the combined blockade of dendritic actin and actomyosin contraction is interesting for understanding macrophage motility modes that are entirely based on formin-containing filopodial protrusions. However, these experiments go beyond the scope of this study, where we focused on the role of integrin-mediated adhesions and substrate-dependent macrophage movement in vitro and in vivo.

5) In the discussion the authors state that: macrophages can "survey their surrounding by two equally efficient strategies: (a) migration-based surveillance by contractile, low-protrusive cells, or (b) protrusion-based surveillance by less contractile, low-migrating cells."The data presented does not support conclusion (a). It can be argued that migration-based surveillance need not be contractile (amoeboid) based, but may also be (and from data appears dominantly to be) driven by highly protrusive and motile, mesenchymal-like motility. In other words, macrophages can survey area through motility (mesenchymal or amoeboid) or non-motile protrusion extension. Again, the use of CK-666 would allow the authors to make a stronger conclusion here.

We agree with this comment and have re-phrased the respective text passages.

B) Are matrigel substrates really 3D as cells appear to settle and migrate in 2D. Are these matrix conditions in which dendritic cells or neutrophil would display much faster chemotaxis as implied in discussion of results? Please expand on methods.

We have now provided more detailed information on this experimental system in the respective Materials and methods section. In brief, the centrifugation step brings the macrophages into the same focal plane for imaging. This step was required because the Incucyte S3 live-cell analysis system operates with an autofocus mode. In this experimental setup the majority of macrophages moves over 24 hours primarily in the lower part of the gel, but cells also move vertically in the 3D gel and leave the autofocus plane. Control experiments with C5a on top of the matrigel recruited all macrophages out of the autofocus plane over 24 h, ensuring the 3D nature of the gel (data not shown). We also ruled out functional interactions between BMDMs and the plastic well bottom. *Itgb2^−/−^* BMDMs, which are nonadherent to cell culture plastic and grow as suspension culture in Petri dishes, show contrasting mesenchymal morphologies in matrigel. Vice-versa, *Itgb1^−/−^* BMDMs, which retain mesenchymal phenotypes and move normally on uncoated Petri dishes due to the promiscuous binding of β2 integrins, mainly αMβ2 (Mac-1), adopt round cell shapes with severely impaired migration in 3D matrigel matrix.